# Specific polar subpopulations of astral microtubules control spindle orientation and symmetric neural stem cell division

Felipe Mora-Bermúdez[1], Fumio Matsuzaki[2], Wieland B Huttner[1]*

[1]Max Planck Institute of Molecular Cell Biology and Genetics, Dresden, Germany; [2]RIKEN Center for Developmental Biology, Kobe, Japan

**Abstract** Mitotic spindle orientation is crucial for symmetric vs asymmetric cell division and depends on astral microtubules. Here, we show that distinct subpopulations of astral microtubules exist, which have differential functions in regulating spindle orientation and division symmetry. Specifically, in polarized stem cells of developing mouse neocortex, astral microtubules reaching the apical and basal cell cortex, but not those reaching the central cell cortex, are more abundant in symmetrically than asymmetrically dividing cells and reduce spindle orientation variability. This promotes symmetric divisions by maintaining an apico-basal cleavage plane. The greater abundance of apical/basal astrals depends on a higher concentration, at the basal cell cortex, of LGN, a known spindle-cell cortex linker. Furthermore, newly developed specific microtubule perturbations that selectively decrease apical/basal astrals recapitulate the symmetric-to-asymmetric division switch and suffice to increase neurogenesis in vivo. Thus, our study identifies a novel link between cell polarity, astral microtubules, and spindle orientation in morphogenesis.

*For correspondence: huttner@ mpi-cbg.de

**Competing interests:** The authors declare that no competing interests exist.

**Reviewing editor**: Freda Miller, The Hospital for Sick Children Research Institute, University of Toronto, Canada

## Introduction

The fundamental functions of the mitotic spindle include not only the faithful partition of the genome into both daughter cells, but also controlling whether cell fate determinants are distributed symmetrically or asymmetrically to those daughters (*Gonczy, 2008*; *Gillies and Cabernard, 2011*). Cell division symmetry is controlled by orienting the metaphase spindle along a specific plane. Cytokinesis then segregates cell components symmetrically or asymmetrically, depending on their distribution on either side of that plane. Pioneering work in fungi and nematodes has shown spindle orientation to involve mitotic astral microtubules. These astrals dynamically link the spindle poles with the cell cortex (*Pearson and Bloom, 2004*; *Siller and Doe, 2009*).

In polarized epithelial cells, the orientation of the mitotic spindle with respect to the apico-basal axis determines the distribution of components located differentially along this axis (*Knoblich, 2008*; *Gillies and Cabernard, 2011*). A classic example is *Drosophila* neurogenesis, where neuroepithelial cells proliferate by dividing symmetrically, with a cleavage plane parallel to the apico-basal axis. Neuroblasts derived from them delaminate from the apical surface and divide in turn asymmetrically, to self-renew and produce neurogenic progenitors. The mitotic spindle in these asymmetric divisions is re-oriented by 90°, with the cleavage plane now perpendicular to the apico-basal axis. This leads to the asymmetric distribution of polarized fate-determinants to the daughter cells (*Southall et al., 2008*; *Sousa-Nunes et al., 2010*). This major spindle re-orientation in *Drosophila* requires interactions between cell cortical Gαi, a heterotrimeric G protein subunit, and Partner of Inscuteable (Pins), which are in turn linked to the Par polarity complex (Par3, Par6, aPKC) by Inscuteable (*Knoblich, 2008*; *Brand and Livesey, 2011*).

Spindle and cleavage plane orientation has also been implicated in the neurogenesis of vertebrates, including mammals (reviewed in *Lancaster and Knoblich, 2012*; *Shitamukai and Matsuzaki*

**eLife digest** A stem cell can divide in two ways. Either it can split symmetrically into two identical daughter stem cells, or it can split asymmetrically into a stem cell and a specialist cell. The structure that forms inside the dividing cell to separate pairs of chromosomes—called the mitotic spindle—also partitions the molecules that determine what kind of cell each daughter cell will become.

The mitotic spindle is made up of protein microtubules. Astral microtubules connect the spindle to a structure found at the inner face of the cell membrane called the cell cortex. This helps the spindle to orient itself correctly and control the plane of cell division. This is particularly important in cells that are different at their top and bottom, like polarized neural stem cells.

To divide symmetrically, these cells need to split vertically from top to bottom. Then, to divide asymmetrically they tilt the cell division plane off-vertical. Classical studies on neuroblasts from the fruit fly *Drosophila* have shown that a big, 90° reorientation, from vertical to horizontal underlies this change. However, in the primary stem cells of the mammalian brain, subtle off-vertical tilting suffices for asymmetric divisions to occur. This tilting must be finely regulated: if not, neurodevelopmental disorders, such as microcephaly and lissencephaly, may arise.

Mora-Bermúdez et al. investigated how mammalian cortical stem cells control such subtle spindle orientation changes by taking images of developing brain tissue from genetically modified mice. These show that not all astral microtubules affect whether the spindle reorients, as was previously thought. Instead, only those connecting the spindle to the cell cortex at the top and bottom of the cell—the apical/basal astrals—are involved.

A decrease in the number of apical/basal astrals enables the spindle to undergo small reorientations. Mora-Bermúdez et al. therefore propose a model in which the spindle becomes less strongly anchored when the number of apical/basal astrals is reduced. This makes the spindle easier to tilt, allowing neural stem cells to undergo asymmetric divisions to produce neurons.

The decrease in the number of apical/basal astrals appears to be caused by a reduction in the amount of a molecule that is known to help link the microtubules to the cell cortex. This reduction occurs only in the cortex at the top of the cell. Mora-Bermúdez et al. were also able to manipulate this process by adding very low doses of a microtubule inhibitor called nocodazole, which reduced the number of only the apical/basal astrals, increasing the ability of the spindle to reorient.

*2012*; see also *Das and Storey, 2012*; *Asami et al., 2011*; *Delaunay et al., 2014*). Mammalian neurogenesis, however, shows major differences to *Drosophila* with regard to spindle orientation in symmetric vs asymmetric divisions of polarized neural stem cells. In the developing neocortex, neuroepithelial cells progressively become radial glia, and both of these highly related subtypes of neural stem cells exhibit a characteristic polarized, apico-basal architecture and undergo apical mitosis, hence the collective term 'apical progenitors' (APs) (*Kriegstein and Götz, 2003*; *Götz and Huttner, 2005*; *Miller and Gauthier, 2007*; *Corbin et al., 2008*; *Martynoga et al., 2012*). Importantly, the switch of APs from symmetric proliferative to asymmetric neurogenic divisions occurs mostly without large and defined re-orientations of the spindle, but with only subtle deviations (*Huttner and Brand, 1997*; *Haydar et al., 2003*; *Kosodo et al., 2004*; *Konno et al., 2008*; *Shitamukai et al., 2011*). These can nonetheless tilt the division plane enough to no longer bisect, but rather bypass the small apical end-foot, leading to its asymmetric distribution (*Kosodo et al., 2004*). Similarly, subtle spindle deviations can also influence the inheritance of the basal process (*Shitamukai et al., 2011*, see also *Kosodo et al., 2008*). The proper regulation of these symmetry changes is thought to be an important determinant in neocortical neurogenesis (*Götz and Huttner, 2005*; *Kriegstein and Alvarez-Buylla, 2009*; *Miyata et al., 2010*), and its perturbation can lead to neurodevelopmental and neurodegenerative disorders (*Feng and Walsh, 2004*; *Fish et al., 2006*; *Yingling et al., 2008*; *Gauthier-Fisher et al., 2009*; *Godin et al., 2010*; *Lizarraga et al., 2010*).

However, it is unknown how the subtle spindle orientation changes that occur in the switch of mammalian APs from symmetric proliferative to asymmetric neurogenic divisions are generated and controlled. Proteins evolutionarily conserved between *Drosophila* and mammals, such as Inscuteable and the Pins homologue LGN, have been shown to be involved in the regulation of AP cleavage plane

orientation and AP daughter cell fate (*Konno et al., 2008*; *Postiglione et al., 2011*). Despite these findings, it remains to be elucidated how alterations in these or other spindle regulators actually lead to the spindle orientation changes that switch the cell division mode of APs (*Buchman and Tsai, 2007*; *Lancaster and Knoblich, 2012*).

Here we show that the spindle orientation that controls symmetric vs asymmetric division of polarized neural stem cells in the developing neocortex is not determined by all astral microtubules, as previously assumed, but by a distinct subpopulation of these microtubules, the apical/basal astrals. Specifically, a decrease in apical/basal astrals causes the subtle spindle orientation deviations that lead to the switch of mammalian APs from symmetric proliferative to asymmetric neurogenic divisions. Moreover, we find that a selective down-regulation of LGN at the basal cell cortex, rather than a general reduction of cell cortical LGN, explains the specific decrease in apical/basal astrals. Our study, which combines live and immunofluorescence microscopy of neocortical tissue from transgenic mice, provides a new model for the control of symmetric vs asymmetric cell division in cells that maintain spindle orientation perpendicular to the apico-basal axis, such as cortical neural stem cells.

## Results

### More dynamic spindle orientation deviations from the apico-basal axis in neurogenic than proliferating progenitors

During metaphase, the orientation of the mitotic spindle of mammalian APs dynamically deviates from the apical, ventricular surface of the developing cortical wall (*Chenn and McConnell, 1995*; *Haydar et al., 2003*; *Roszko et al., 2006*). Crucially however, whether such deviations are different in proliferating vs neurogenic progenitors is unknown. To ask if the amplitude of spindle deviations is involved in neurogenesis, a transgenic mouse was used where neurogenic divisions are identified by EGFP driven by the promoter of the pan-neurogenic marker Tis21 (*Iacopetti et al., 1999*; *Haubensak et al., 2004*) and thereby distinguished from proliferative divisions. Specifically, embryonic day (E) 14.5 mouse dorsolateral telencephalon in organotypic slice culture stained with a vital DNA dye was analysed by dual-colour live 3D imaging. Consistent with previous studies, the metaphase spindles of most progenitors, as reported by the chromosome plates, continuously changed their angles with respect to the apical–basal axis until anaphase, and did so in apparently random fashion. Interestingly, these dynamic deviations increased in amplitude from Tis21::GFP– (proliferating) AP divisions to Tis21::GFP+ (neurogenic) AP divisions (*Figure 1A,B,D,E*; *Videos 1,2*). In neurogenic APs, the total amplitude of deviations was on average twice as high as in proliferating APs (*Figure 1G*; AP proliferating 15.7 ± 1.4°, AP neurogenic 30.7 ± 4.3°). Importantly, the range of metaphase spindle orientations (*Figure 1D,E*) was consistent with the previously reported cleavage plane orientations of fixed (*Kosodo et al., 2004*) and live (*Konno et al., 2008*) mouse APs, which have been observed to be mostly perpendicular to the apical surface.

Spindle orientation variability was also measured in basal progenitors (BPs), which originate from APs but are delaminated from the ventricle and divide basally. Mouse BPs also lack the canonical apical–basal polarity features of APs (*Attardo et al., 2008*) and mostly undergo terminal divisions that produce two neurons (*Haubensak et al., 2004*; *Miyata et al., 2004*; *Noctor et al., 2004*). Compared to proliferating APs, neurogenic BPs showed deviations of the metaphase plate axis from the apical–basal axis of the developing cortical wall that were even greater than those of neurogenic APs (*Figure 1C,F*; *Video 3*). The total amplitude of deviations in BPs increased to 38.3 ± 5.6° (*Figure 1G*). Importantly, the range of metaphase spindle orientations (*Figure 1F*) was consistent with the highly variable cleavage plane orientation of mouse BPs reported previously (*Haubensak et al., 2004*; *Attardo et al., 2008*). This leads to the conclusion that metaphase spindle orientation variability increases as progenitors become neurogenic. Interestingly, this variability is characteristic of both, neurogenic APs that divide asymmetrically with a cleavage plane largely parallel to the apical–basal tissue axis, and of neurogenic BPs which divide symmetrically with a near-random cleavage plane orientation.

### Neurogenic progenitors have fewer apical and basal astral microtubules

To investigate the causes of increased dynamic spindle deviations in neurogenic progenitors, the mitotic microtubules were analysed (*Figure 2A–F*). Changes in spindle dynamics could be caused by different numbers of astral microtubules interacting with the cell cortex. Interestingly, the number of astrals reaching the cell cortex was highest in proliferating APs (*Figure 2G*; 21.65 ± 1.07) followed by neurogenic APs (16.70 ± 0.98), and lowest in neurogenic BPs (11.36 ± 0.49).

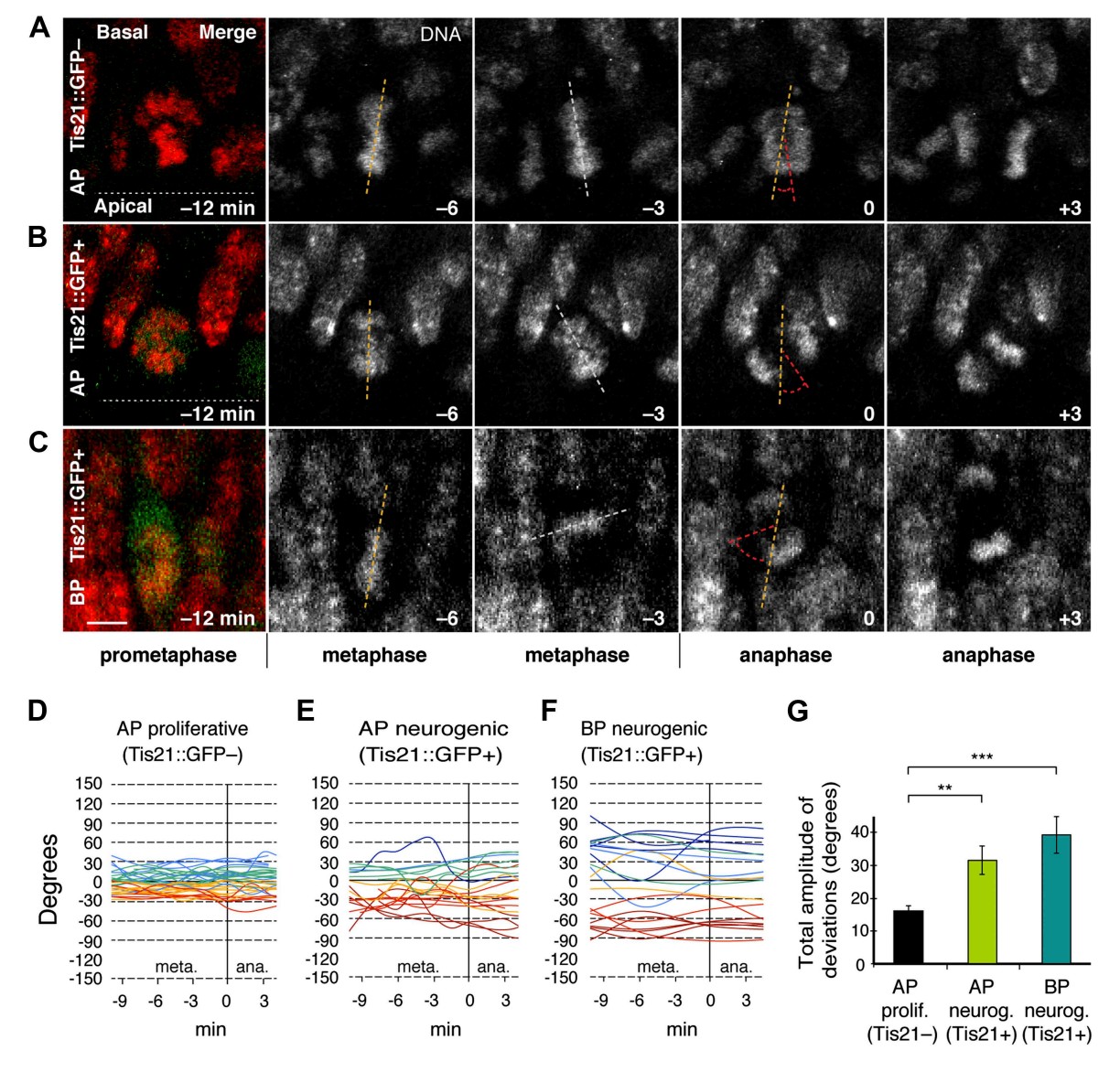

**Figure 1**. Dynamic spindle orientation variability increases when progenitors become neurogenic. Live tissue imaging of spindle orientation as reported by chromosome plate orientation in organotypic slice culture of coronal sections from E14.5 Tis21::GFP mouse dorsolateral telencephalon. Observations focused on metaphase, when all chromosomes congressed to the equatorial plane and the dynamics observed were likely more related to spindle orientation, rather than the prometaphase establishment of a functional spindle and chromosome plate. 0 min is anaphase onset. (**A**–**C**) Apical progenitor (AP; **A**, **B**) and basal progenitor (BP; **C**) undergoing either proliferative (**A**, Tis21::GFP–) or neurogenic (**B**, **C**; Tis21::GFP+) division. Merge: chromosomes in red, EGFP in green. Chromosome plate orientation was determined by measuring angular deviations from the apico-basal axis (0°, corresponding to vertical), which runs perpendicular to the apical ventricular surface (90°, horizontal white dotted lines in **A**, **B**). Time-lapse is 3 min. Vertical or oblique dashed lines indicate chromosome plate orientations in metaphase or anaphase onset; maximal deviation angles for each plate were set by the orientation at an early time-point (yellow) and a later time-point (red), which are quantified in **G**. White dashed lines indicate intermediate orientations. Scale bar = 5 μm. (**D**–**F**) Quantification of chromosome plate orientations from metaphase (meta.) to anaphase (ana.). To facilitate tracing, individual tracks are colour-coded according to the range in which most of the track remained (blue, beyond 30°, cyan 30°–15°, green 15°–0°, yellow 0° to −15°, red −15° to −30°, dark red, beyond −30°). (**G**) Mean ± standard error of the mean (SEM) of the maximal amplitude of deviations for proliferative (prolif.) and neurogenic (neurog.) divisions. **p<0.001, ***p<0.0001, Kruskal–Wallis ANOVA (K–W) with Dunn's multiple comparison (DMC) test; n = 20 progenitors per category, from three independent litters and experiments. See also *Videos 1–3*.

The next question was whether this decrease was spatially homogeneous around the cell. Considering the highly polarized architecture of APs (*Götz and Huttner, 2005*; *Mora-Bermúdez et al., 2013*), the progenitor soma was divided into three regions, to count the astrals reaching the cell

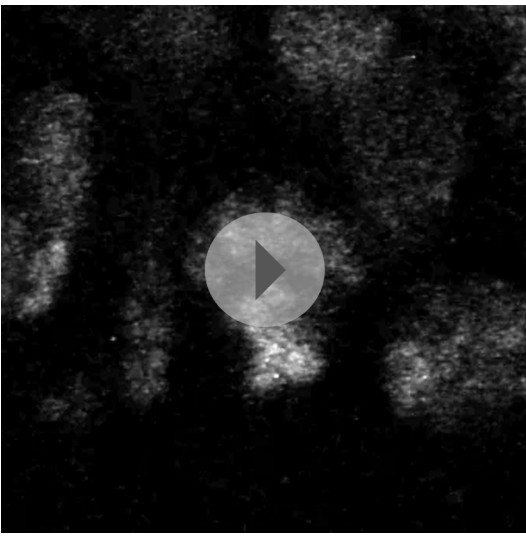

**Video 1**. Dynamic spindle orientation variability in a representative proliferating AP. Related to **Figure 1A**. Live tissue imaging of spindle orientation, as reported by the chromosome plate (DNA) orientation, in organotypic slice culture of dorsolateral telencephalon coronal sections, from an E14.5 Tis21::GFP mouse. Time-lapse 3 min. Total time elapsed 18 min.

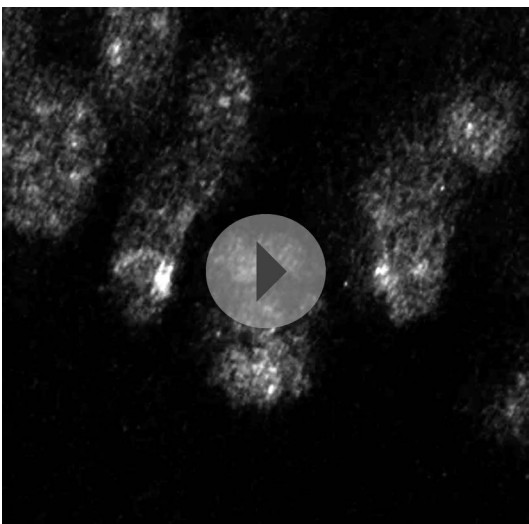

**Video 2**. Dynamic spindle orientation variability in a representative neurogenic AP. Related to **Figure 1B**. Live tissue imaging of spindle orientation, as reported by the chromosome plate (DNA) orientation, in organotypic slice culture of dorsolateral telencephalon coronal sections, from an E14.5 Tis21::GFP mouse. Time-lapse 3 min. Total time elapsed 18 min.

cortex in each (**Figure 2A** and 'Materials and methods'): the central region, defined by the presence of the centromeres and containing most of the spindle—including some astrals— and an apical and a basal region, both containing only astrals. Surprisingly, the decrease in astrals was specific to the apical and basal regions. The number of astrals reaching the apical or basal region of the cell cortex (apical/basal astrals) was lower in neurogenic than in proliferating APs (**Figure 2H**; neurogenic 7.2 ± 0.36, proliferating 11.8 ± 0.43). By contrast, the number of astrals reaching the central region of the cell cortex (central astrals) did not change significantly (**Figure 2I**; neurogenic APs 8.8 ± 0.7, proliferating APs 9.1 ± 0.6).

To compare APs and BPs, the apical, basal, and central populations of astrals from APs were compared to the apically, basally, and centrally oriented populations of astrals in BPs (see 'Materials and methods'). In contrast to APs, BPs showed a general decrease, with fewer astrals than APs for both populations (**Figure 2H,I**, apically/basally oriented 5.5 ± 0.28, centrally oriented: 6.1 ± 0.46). Together with our data on spindle orientation variability (**Figure 1**), this suggests that a higher abundance of apical/basal astrals increases the stability of spindle orientation.

The numbers of basal and apical astrals were then compared. Surprisingly, in proliferating APs, basal astrals were more abundant than apical ones (**Figure 2J**, 6.7 ± 0.31 vs 5.1 ± 0.25). By contrast, in neurogenic APs, the numbers of both basal and apical astrals were lower than in proliferating APs, but were not significantly different from each other (basal 3.7 ± 0.2, apical 3.4 ± 0.2). This shows that the number of basal astrals is initially higher in proliferating APs, but their decrease is such that neurogenic APs have a homogeneously low number of apical and basal astrals. This trend continued in BPs, where the number of apically and basally oriented astrals further decreased homogeneously (basally oriented 2.7 ± 0.2, apically oriented 2.6 ± 0.2).

## Fewer apical/basal astral microtubules as neurogenesis progresses

In mammals, the proportion of neurogenic divisions of APs increases throughout embryonic neurogenesis, which in the mouse goes from around E10 to E17. Therefore, the total number of apical/ basal astrals could decrease as neurogenesis progressed, from E11.5 to E14.5 to E16.5 (early, mid, and late neurogenesis, respectively). Indeed, the number of astrals in APs decreased 2.5-fold from early to late neurogenesis, from 15.2 ± 0.4 at E11.5 to 9.4 ± 0.5 at E14.5, and further to 6.2 ± 0.3 at E16.5 (**Figure 3**). This provides further evidence that

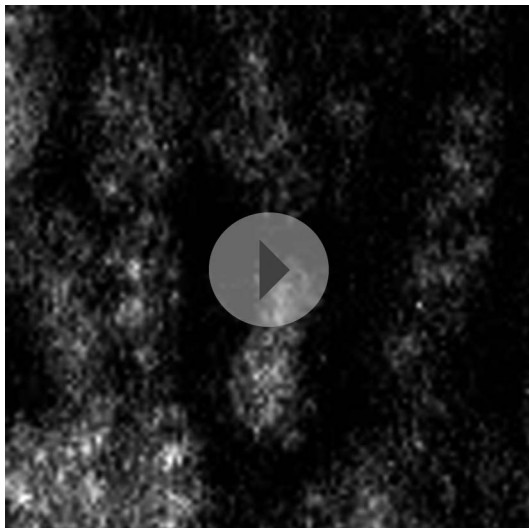

**Video 3**. Dynamic spindle orientation variability in a representative neurogenic BP. Related to *Figure 1C*. Live tissue imaging of spindle orientation, as reported by the chromosome plate (DNA) orientation, in organotypic slice culture of dorsolateral telencephalon coronal sections, from an E14.5 Tis21::GFP mouse. Time-lapse 3 min. Total time elapsed 18 min.

dividing neurogenic APs have fewer apical/basal astrals than proliferating ones and suggests a role for these astrals in the switch to neurogenesis.

## Fewer apical/basal astral microtubules in APs of LGN KO mice and APs overexpressing dominant-negative LGN-C

Previous studies have shown LGN to be a key player in spindle orientation. LGN is enriched in the cell cortex, especially around the central part of the cell, where it participates in spindle orientation in non-neural cells in vitro (*Du and Macara, 2004*). Also, upon LGN perturbation in neural progenitors in vivo, spindle orientation variability increased, changing progenitor identity (*Morin et al., 2007*; *Konno et al., 2008*; *Shitamukai et al., 2011*). How LGN regulates spindle orientation in neurogenesis remains unknown. LGN could control spindle orientation in neural progenitors by regulating the number of astrals that anchor the spindle to the cell cortex. Therefore, the higher spindle orientation variability observed upon LGN perturbation could be caused by fewer apical/basal astrals. Indeed, in the E14.5 neocortex of LGN knock-out (KO) mice (*Konno et al., 2008*) (*Figure 4A–D*), fewer apical/basal astrals were found in APs of homozygous compared to heterozygous mice (*Figure 4E*; homozygous 8.4 ± 0.5, heterozygous 11.2 ± 0.7). By contrast, no significant reduction was seen in the central astrals (*Figure 4F*; homozygous 9.1 ± 0.5, heterozygous 8.9 ± 0.3, similar to unperturbed APs; *Figure 2I*). No significant change was observed in BPs either (*Figure 4G*; homozygous 5.9 ± 0.4, heterozygous 6.4 ± 0.4), where the number of apically/basally oriented astrals was also similar to unperturbed BPs (*Figure 2H*).

To corroborate and complement these results, in utero electroporation was used to overexpress the dominant-negative C-terminal region of LGN (LGN-C) in the dorsolateral telencephalon of wt mice. LGN-C has the GoLoco motifs that bind to Gαi at the cell cortex, and its overexpression saturates the Gαi anchoring sites for microtubules. However, it lacks the TPR motifs in the N-terminal region and cannot interact with microtubules via NuMA and dynein (*Morin et al., 2007*; *Shitamukai et al., 2011*). APs overexpressing 6Myc-LGN-wt or 6Myc-LGN-C showed staining mostly around the basolateral cell cortex (*Figure 4I,J*), similar to previous LGN stainings (*Konno et al., 2008*). Consistent with our LGN-KO mouse results, APs overexpressing LGN-C also showed a significant reduction in apical/basal astrals at E14.5 (*Figure 4J,K*, 7.9 ± 0.5), similar to LGN KO homozygous mouse APs (*Figure 4E*). By contrast, APs in the electroporated area that were either non-electroporated or overexpressing LGN-wt had a similar number of apical/basal astrals (*Figure 4H,I,K*, 11.0 ± 0.6 and 11.2 ± 0.4) as unperturbed (*Figure 2H*) or LGN-KO heterozygous (*Figure 4E*) APs. Together, these results show that LGN perturbations lead to a reduction specifically in apical/basal astrals, likely through reduced anchoring of astrals to the cell cortex.

## Cell-cortical LGN localization changes between progenitor types and corresponds with the astral microtubule abundance pattern

The higher spindle orientation stability in proliferating APs could depend on cortical LGN levels. To test this, the distribution of LGN was analysed in different progenitors. No major differences in overall levels of LGN immunoreactivity were observed, but the subcellular distribution of the signal was different between proliferating and neurogenic E14.5 APs (*Figure 5A–C*). In proliferating APs, cortical LGN increased from the apical to the central region of the cell cortex and remained high in most of the central and basal regions (*Figure 5A,E*). In neurogenic APs, cortical LGN levels were similar to proliferating APs at the apical and central regions, but markedly lower at the basal region (*Figure 5B,E*).

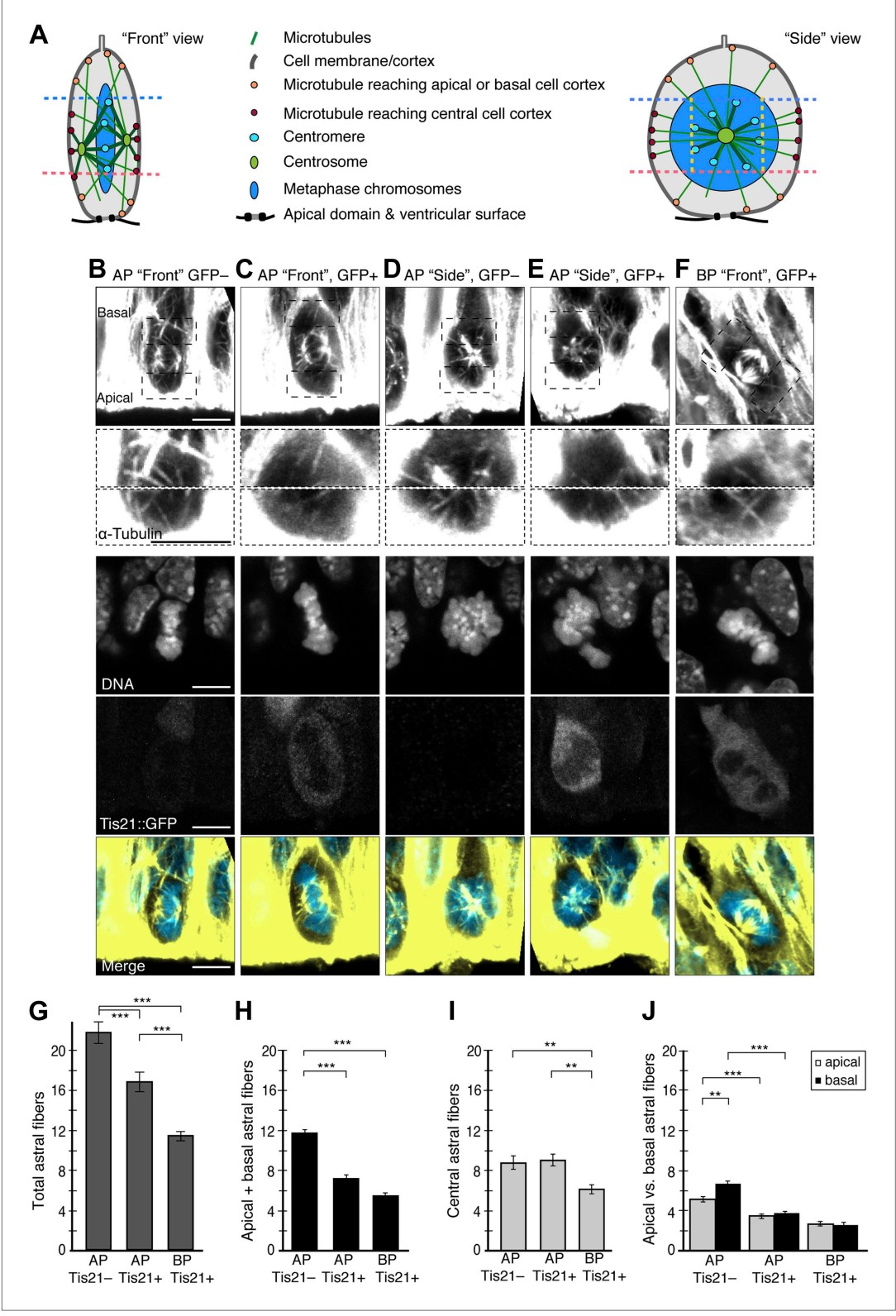

**Figure 2**. Fewer astral microtubules when progenitors become neurogenic. (**A**) Cartoons of the mitotic spindle in APs as appearing upon 'Front' and 'Side' viewing of the same chromosome plate, rotating around its apico-basal axis. Apical- and basal-reaching astrals (apical/basal astrals) were defined as those extending beyond the main chromosomal area, delimited by the apical-most (pink dashed line) and basal-most (blue dashed line) peri/centromeric

*Figure 2. Continued on next page*

*Figure 2. Continued*

heterochromatin foci (brightest points in DNA, see **B–F**), and reaching either the apical or basal region of the cell cortex, which are delimited by these same pink and blue dashed lines, respectively. In the 'Side' view, yellow dashed lines delimit the peri/centromeric heterochromatin region laterally, beyond which the astrals defined as central-reaching (central astrals) extend. (**B–F**) α-tubulin immunofluorescence (maximum intensity projections of two 0.75 μm optical sections) of E14.5 Tis21::GFP mouse dorsolateral telencephalon showing mitotic microtubules in proliferating APs (Tis21::GFP–) vs neurogenic APs (Tis21::GFP+) vs neurogenic BPs (Tis21::GFP+). Dashed boxes show regions including the basal and apical cell cortex that are shown in the second and third row, respectively, at higher magnification and brightness. DNA staining (DAPI) and Tis21::GFP fluorescence are single 0.75 μm optical sections. Merge: DNA in blue, microtubules in yellow. Scale bars = 5 μm. (**G–J**) Mean numbers per cell of astrals reaching the cell periphery: all astrals (**G**), apical/basal astrals (**H**), central astrals (**I**), and apical or basal astrals (**J**). (**G, H, J**) In BPs, astrals are considered as apically, basally, and centrally oriented (see 'Materials and methods'). n = 40 progenitors per category and **I** n = 20 progenitors per category, all from 8 independent litters and experiments. *p<0.05, **p<0.001, ***p<0.0001; K–W with DMC post test. Error bars are SEM.

These data show that neurogenic APs lose LGN enrichment in the basal region of the cell cortex, which could lead to less anchoring of basal astrals. Consistent with this, the strongest decrease in astrals of APs was in the basal region of neurogenic progenitors (*Figure 2J*). A more modest but significant decrease in astrals in the apical region was also observed, yet the levels of apical LGN remained low in all progenitors. This suggests that additional factors may contribute to spindle orientation stability at the apical side.

Furthermore, compared to APs, cortical LGN levels in BPs were much lower, distributed more homogeneously over the cell and with only minor cortical enrichments (*Figure 5C,E*). This is consistent with BPs having the lowest overall numbers of apical/basal and central astrals (*Figure 2G–I*), as well as the highest amplitudes of dynamic spindle deviations (*Figure 1F,G*). Together, these results strongly support a regulation of astral microtubule abundance by an LGN-dependent anchoring that depends on the specific cortical enrichment of LGN.

## Changing the number of apical/basal astral microtubules in APs recapitulates the transition between proliferating and neurogenic divisions

### Astral microtubule abundance

If more apical/basal astrals are needed to maintain symmetric proliferative AP divisions, reducing their number could switch APs from proliferation to neurogenesis by increasing asymmetric neurogenic divisions. To test this, Brain Rotation Organotypic (BRO) culture (see 'Materials and methods') of whole E14.5 forebrains was performed with the microtubule polymerization inhibitor nocodazole. Nocodazole was titrated to lower the number of astrals in proliferating APs, but not lower than the number found in unperturbed neurogenic APs (*Figure 6A–E*). In the presence of only solvent (DMSO), the mean number of apical/basal astrals in proliferating APs was 10.9 ± 0.4 (*Figure 6E*), similar to untreated proliferating APs (*Figure 2H*). With extremely low (20–40 pM) concentrations of nocodazole, a gradual reduction in the number of these apical/basal astrals was observed (*Figure 6A–E*). Most interestingly, in the presence of 30 pM nocodazole for 2.5 hr, the mean number of apical/basal astrals in proliferating APs was reduced to 6.7 ± 0.4 (*Figure 6E*), similar to untreated neurogenic APs (*Figure 2H*).

It was necessary to ensure that incubation with 30 pM nocodazole (referred to as minimal nocodazole) significantly affected only the apical/basal astrals of proliferating APs, and no other mitotic microtubules nor microtubule-based structures of interphase cells. For this, we observed that, first, neither the number of central astrals in proliferating APs, nor the number of central or apical/basal astrals in neurogenic APs, was reduced with 30 pM nocodazole (*Figure 6E*). Second, key spindle parameters, such as its area and the amount and distribution of microtubules, were not significantly affected (*Figure 7A–D* and 'Materials and methods'). Third, relevant cell and tissue features were indistinguishable between controls and minimal nocodazole incubations, namely (i) cell diameter (*Figure 7E,F*); (ii) the morphology of primary cilia, an interphase microtubule-dependent cell protrusion; (iii) the elongated bipolar morphology of APs; and (iv) the overall cytoarchitecture of the cortical wall as revealed by the pattern of microtubules (*Figure 7—figure supplement 1A–H*, respectively). This shows that minimal nocodazole treatment recapitulates the selective reduction in apical/basal astrals

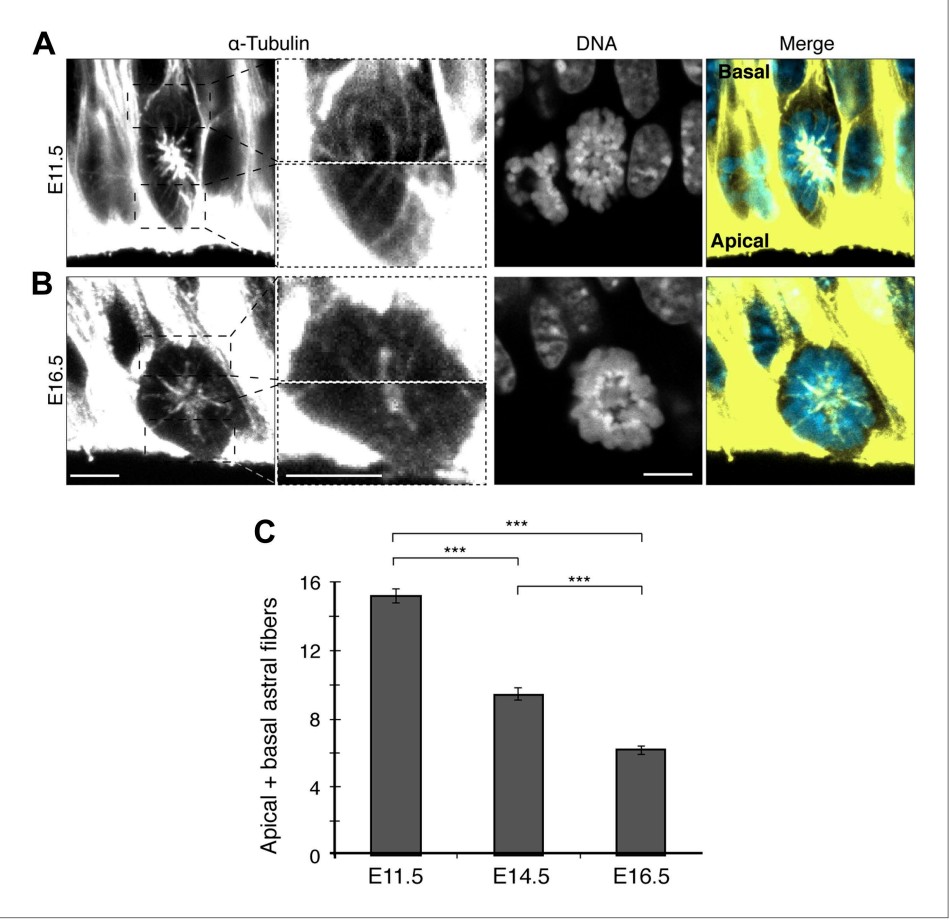

**Figure 3**. Fewer apical/basal astral microtubules in progenitors as neurogenesis progresses. **A**, **B**) α-Tubulin immunofluorescence (maximum intensity projections of two 0.75 µm optical sections) of E11.5 and E16.5 mouse dorsolateral telencephalon showing mitotic microtubules in APs. Dashed boxes in the first column show regions including the basal and apical cell cortex that are shown at higher magnification and brightness in the second column. DNA stainings (DAPI) are single 0.75 µm optical sections. Merge: DNA in blue, microtubules in yellow. Scale bars = 5 µm. (**C**) Mean number per cell of apical/basal astrals in APs at E11.5, E14.5 (*Figure 2H*, mean of both AP categories), and E16.5. ***p<0.0001, K–W with DMC post test; for E11.5 and E16.5, n = 20 APs, from 4 independent litters and experiments; Error bars are SEM.

observed in the transition of unperturbed proliferating APs to neurogenic APs, without affecting important cell and tissue features (see below for unaffected mitotic progression).

Conversely, an increase in the number of apical/basal astrals was pursued in neurogenic APs to the level found in proliferating APs, but not higher. For this, BRO culture of E14.5 Tis21::GFP forebrains was performed for 2.5 hr with the microtubule stabilizer taxol (*Figure 6F–H*). The extremely low taxol concentration of 25 pM increased the mean number of apical/basal astrals in neurogenic APs to 10.5 ± 0.6 (*Figure 6H*), similar to DMSO-only treated and untreated proliferating APs (*Figure 6E*, *Figure 2H*). Importantly, the number of apical/basal astrals in proliferating APs under 25 pM taxol was not significantly higher (*Figure 6H*, 11.1 ± 0.5), showing that this increase is specific for neurogenic APs.

Next, the question was whether combining taxol treatment with overexpression of wt LGN, a microtubule-cortex linker, would further increase the number of apical/basal astrals. For technical reasons (see 'Materials and methods'), this was performed in wt forebrains, where the mixed subpopulations of proliferating and neurogenic APs could not be distinguished from each other (*Figure 6I–K*). Indeed, in the presence of 25 pM taxol, the mean number of apical/basal astrals in APs increased further to 12.3 ± 0.7, significantly higher than wt LGN-overexpressing APs incubated only with DMSO

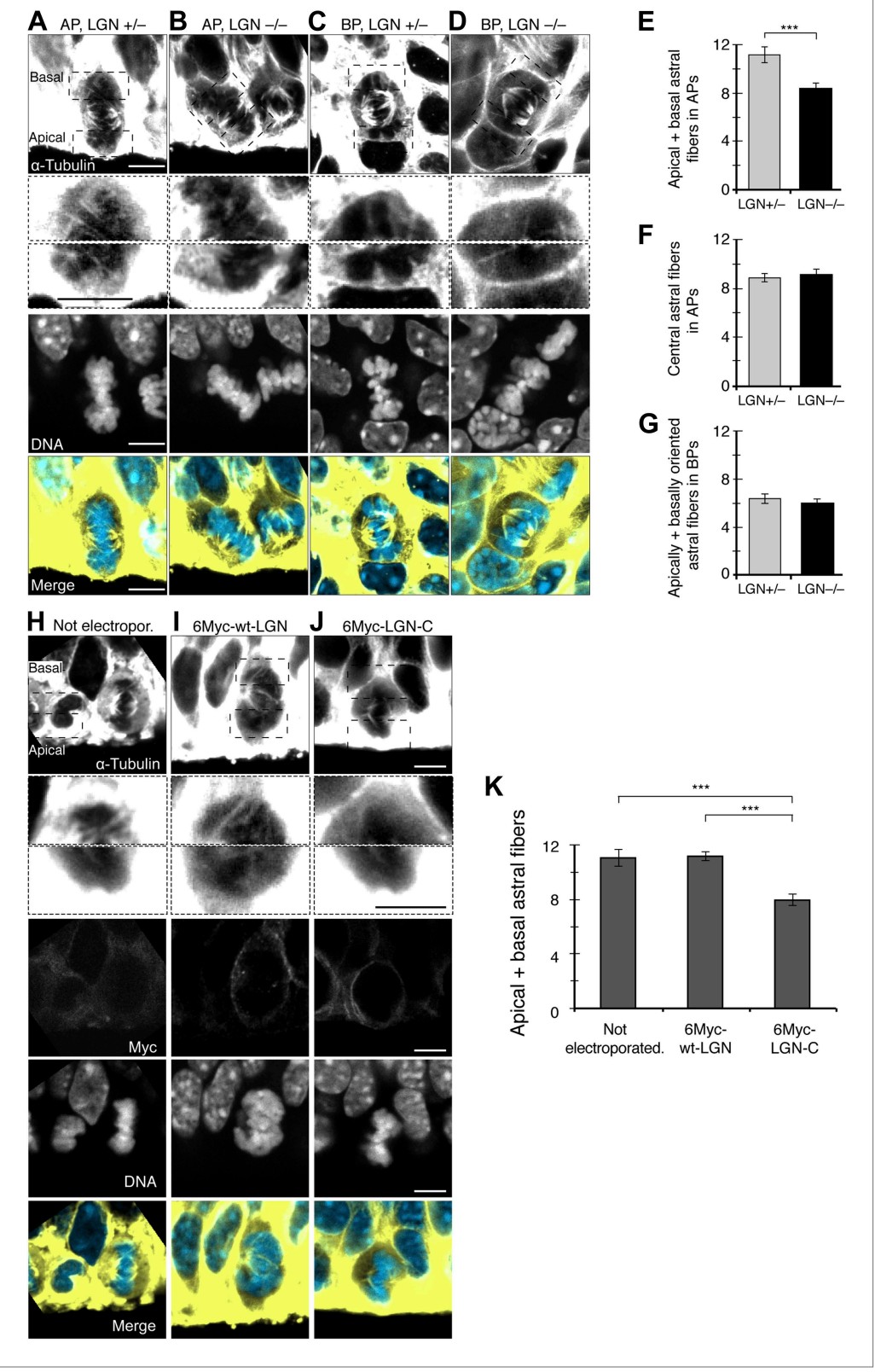

**Figure 4**. Fewer astral microtubules in LGN KO progenitors and upon dominant-negative LGN-C overexpression. (**A–D**) α-Tubulin immunofluorescence (maximum intensity projections of two 0.75 μm optical sections) of E14.5 heterozygous (LGN +/−) vs homozygous (LGN −/−) E14.5 LGN KO mouse dorsolateral telencephalon showing
*Figure 4. Continued on next page*

*Figure 4. Continued*

mitotic microtubules in APs and BPs. Dashed boxes show regions including the basal and apical cell cortex that are shown in the second and third row, respectively, at higher magnification and brightness. DNA staining (DAPI) and Tis21::GFP fluorescence are single 0.75 µm optical sections. Merge: DNA in blue, microtubules in yellow. Scale bars = 5 µm. (**E**) Mean number per AP of astrals reaching the apical and basal cell cortex (apical/basal astrals). (**F**) Mean number per AP of astrals reaching the central cell periphery (central astrals). (**G**) Mean number per BP of astrals that are apically and basally oriented. ***p=0.0007, one-tail *t*-test, n (20 progenitors per category from 3 independent litters and experiments). Error bars are SEM. See also *Figure 5*. (**H–J**) α-Tubulin immunofluorescence (maximum intensity projections of two 0.75 µm optical sections) of in utero electroporated E13.5 wt mouse dorsolateral telencephalon analysed 24 hr later (E14.5). APs in the electroporated regions were either not electroporated (**H**), or electroporated with LGN-wt (6Myc-wt-LGN, **I**) or with dominant-negative LGN (6Myc-LGN-C, **J**). Dashed boxes in the first column show regions including the basal and apical cell cortex that are shown at higher magnification and brightness in the second column. DNA staining (DAPI) and Myc immunofluorescence are single 0.75 µm optical sections. Merge: DNA in blue, microtubules in yellow. Scale bars = 5 µm. (**K**) Mean number per cell of astral microtubules reaching the apical or basal cell cortex (apical + basal astrals) of APs in the electroporated region that were either not electroporated, or transfected with 6Myc-LGN-wt or with 6Myc-LGN-C. ***p<0.0001, one-way ANOVA with TMC test; n = 20 progenitors per category, from 3 independent litters and experiments. Error bars are SEM. Scale bars = 5 µm.

(9.9 ± 0.6) (*Figure 6K*). This shows that a combination of taxol treatment and wt LGN overexpression can also increase the number of apical/basal astrals in APs.

Taxol incubation could have a similar effect in BPs. However, no significant change was detected in apically and basally oriented astrals in BPs electroporated with LGN-wt, that were incubated either with DMSO-only or 25 pM taxol (*Figure 6—figure supplement 1A–C*; Control 6.5 ± 0.6, taxol 7.4 ± 0.4).

## Spindle orientation variability

In light of these observations, it was important to determine whether changing the number of apical/basal astrals could affect spindle orientation variability in proliferating APs. First, live imaging of organotypic slice culture was used to test whether an increase in astrals mediated by taxol could decrease spindle orientation variability in APs (*Figure 8A–D*; *Videos 4–7*). Interestingly, 25 pM taxol alone did not significantly decrease the amplitude of spindle orientation variability, neither in proliferating nor in neurogenic APs (*Figure 8G*; DMSO proliferating 13.3 ± 1.7°, taxol proliferating 14.2 ± 1.6°, DMSO neurogenic 22.5 ± 2.7°, taxol neurogenic 20.4 ± 2.1°).

To reduce spindle orientation variability, taxol-stabilized microtubules could require higher than physiological levels of LGN at the cell cortex. For technical reasons (see 'Materials and methods'), taxol treatment combined with in utero electroporation of LGN-wt for overexpression was performed in wt forebrains. Proliferating and neurogenic APs could therefore not be distinguished from each other in the live imaging of organotypic slice cultures. In this population of mixed APs (*Figure 8E–F*, *Videos 8,9*), spindle orientation variability decreased to a level (15.1 ± 2.3°) similar to control, non-LGN-overexpressing proliferating APs (*Figure 8G*). In DMSO-only treated mixed APs overexpressing LGN-wt, spindle orientation variability was at an intermediate level (18.1 ± 2.1°) between control proliferating and neurogenic progenitors, consistent with a mixed AP population that remains unchanged under control conditions.

Interestingly, no such change was detected in BPs (*Figure 8—figure supplement 1A–C*; control 44.2 ± 9.5°, taxol 41.7 ± 11.0°). This suggests that the stabilization of spindle orientation variability produced by an increase in apical/basal astrals is specific of APs. This difference is not fully unexpected. BPs are substantially different to APs, as they have delaminated from the apical surface and therefore lack canonical apical–basal polarity features of APs (*Attardo et al., 2008*). A consequence of this polarity loss is that, regardless of their mitotic cleavage orientation, most mouse BPs undergo terminal divisions, which are symmetric in that two neurons are produced (although the neurons could be of different type) (*Haubensak et al., 2004*; *Miyata et al., 2004*; *Noctor et al., 2004*). Therefore, changes in spindle orientation are not predicted to significantly change the symmetry or outcome of BP divisions. Consistent with our results, it is thus plausible that BPs may naturally be less responsive to the microtubule and spindle changes that are here described as key for APs.

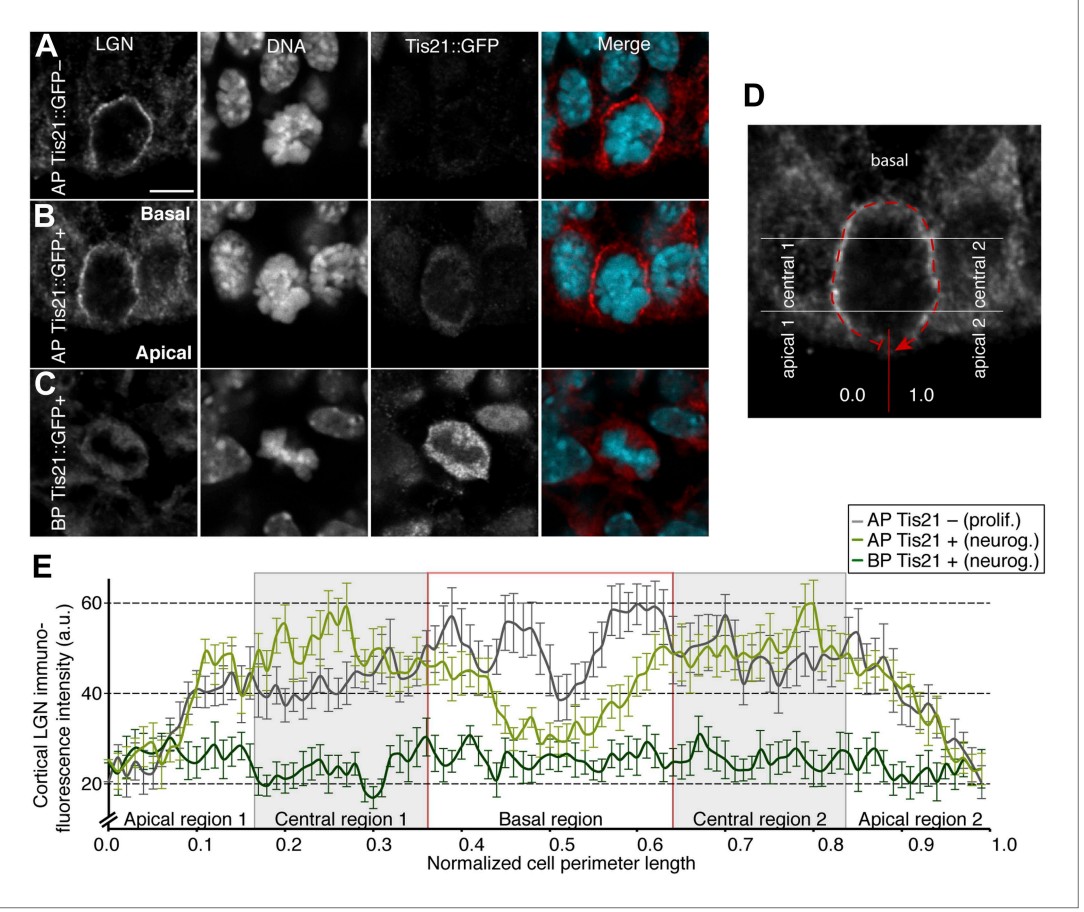

**Figure 5**. Less cell cortical LGN in neurogenic progenitors. (**A–C**) Double immunofluorescence for LGN and GFP, with DNA staining (DAPI), of E14.5 Tis21::GFP mouse dorsolateral telencephalon showing representative examples of a proliferating AP (Tis21::GFP–) vs neurogenic AP (Tis21-GFP+) vs neurogenic BP (Tis21-GFP+). Merge: DNA in cyan, LGN in red. (**D**) Same mitotic AP as in (I), illustrating how LGN cortical immunofluorescence intensity was measured, starting at 0.0 in the middle of the apical region and continuing clockwise along the entire cell cortex (dashed red line) until 1.0. Apical, central, and basal regions are indicated (regions 1, left side; regions 2, right side). (**E**) Mean ± SEM of LGN cortical immunofluorescence intensity along 100 equidistant points of the normalized cell perimeter length; the light grey boxes highlight the central regions; the red box highlights the basal region; n = 15 progenitors per category, from 4 independent litters and experiments. Scale bars = 5 μm.

Together, these results show that, whereas in APs taxol alone is sufficient to increase the number of apical/basal astrals, it requires a concomitant overexpression of functional LGN to decrease spindle orientation variability and thus revert to a more proliferation-like mitotic behaviour. This is consistent with a model where a simple increase in the sub-population of apical/basal astrals is not enough and these astrals must be anchored to the cell cortex in order to exert a stabilizing effect on spindle orientation.

Conversely, a decrease in apical/basal astrals mediated by minimal nocodazole could lower spindle orientation variability in APs. Interestingly, using live imaging of organotypic slice culture, incubation with 30 pM nocodazole was sufficient to increase the amplitude of spindle orientation variability in proliferating APs to a level similar to control neurogenic APs (*Figure 9A–E*; *Videos 10–13*; Control proliferating 13.1 ± 1.7°, nocodazole proliferating 23.8 ± 3.8°, Control neurogenic 22.7 ± 2.7°). The amplitude in neurogenic APs under minimal nocodazole did not increase significantly (25.5 ± 4.1°), showing that this nocodazole concentration only changes the spindle orientation variability of proliferating APs.

## Asymmetric cell division with respect to the apical domain

It was important to determine if a reduction in the number of apical/basal astrals with 30 pM nocodazole could also increase AP division asymmetry. An established assay was adapted to determine the proportion

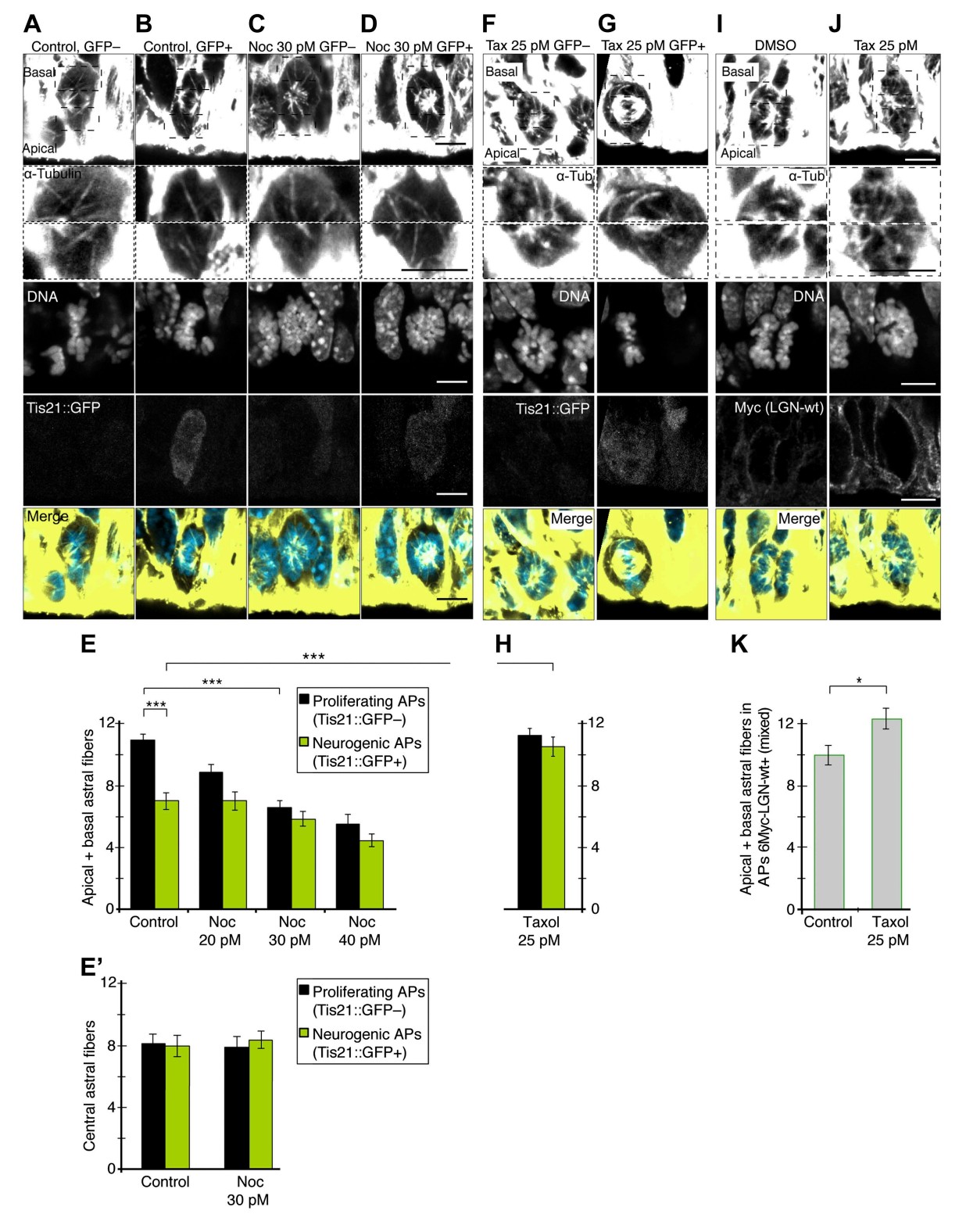

**Figure 6**. Changing the number of apical/basal astral microtubules in APs. **A–E**) Controlled decrease in the apical/basal astrals of proliferating APs under minimal nocodazole. E14.5 forebrains from Tis21::GFP mice were incubated for 2.5 hr in BRO culture, either with solvent (DMSO) only (Control) or with 30 pM nocodazole (Noc). (**A–D**) α-Tubulin immunofluorescence (maximum intensity projections of two 0.75 μm optical sections) of dorsolateral

*Figure 6. Continued on next page*

*Figure 6. Continued*

telencephalon showing mitotic microtubules in proliferating APs (Tis21::GFP–) vs neurogenic APs (Tis21::GFP+), either in controls or with 30 pM nocodazole. Dashed boxes show regions including the basal and apical cell cortex that are shown in the second and third row, respectively, at higher magnification and brightness. DNA staining (DAPI) and Tis21::GFP fluorescence are single 0.75 µm optical sections. Merge: DNA in blue, microtubules in yellow. Scale bars = 5 µm. (**E**) Mean number per cell of apical/basal astrals in control or upon incubation with increasing concentrations of nocodazole in the picomolar range. Note that, with 30 pM nocodazole, the mean number of apical/basal astrals of proliferating APs (Tis21::GFP–) was both significantly lower than in control proliferating APs and similar to that of control neurogenic APs (Tis21::GFP+); ***$p < 0.0001$, K–W with DMC post test; n = 20 progenitors per category, from 3 independent litters and experiments. (**E'**) Mean number per cell of central astrals in controls or with 30 pM nocodazole. n = 10 progenitors per category, from 3 independent litters and experiments. Error bars are SEM. See also *Figure 6—figure supplement 1*. (**F–K**) Controlled increase in apical/basal astrals of neurogenic APs under minimal taxol and also in APs overexpressing LGN-wt combined with minimal taxol. (**F**, **G**) E14.5 forebrains from Tis21::GFP mice were incubated for 2.5 hr in BRO culture with 25 pM taxol. α-Tubulin immunofluorescence (maximum intensity projections of two 0.75 µm optical sections) of dorsolateral telencephalon showing mitotic microtubules in a proliferating AP (Tis21::GFP–) vs a neurogenic AP (Tis21::GFP+). Dashed boxes in the first row show regions including the basal and apical cell cortex that are shown at higher magnification and brightness in the second and third rows. Tis21::GFP fluorescence (fifth row) and DNA staining (DAPI, fourth row) are single 0.75 µm optical sections. Merge: DNA in blue, microtubules in yellow. Scale bars = 5 µm. (**H**) Mean number per cell of apical/basal astrals upon 25 pM taxol incubation. The mean number of apical/basal astrals of neurogenic APs (Tis21::GFP+) was both significantly higher than in control neurogenic APs (**E**) and similar to that of control (**E**) and taxol-treated (**H**) proliferating APs (Tis21::GFP–). n = 12 progenitors per category, from three independent litters and experiments. (**I**, **J**) E13.5 wt forebrains were subjected to in utero electroporation of 6Myc-LGN-wt and 18 hr later (E14.25) incubated for 2.5 hr in BRO culture with either DMSO only (control) or 25 pM taxol. Immunofluorescences similar to **F**, **G**, but the fifth row shows immunofluorescence for Myc (6Myc-LGN-wt). (**K**) Mean number of apical/basal astrals of APs electroporated with 6Myc-LGN-wt. This was higher upon taxol incubation than in controls. *$p < 0.05$, ***$p < 0.0001$, K–W with DMC post test; n = 12 progenitors per category, from n ≥ 2 independent litters and experiments. Error bars are SEM.

The following figure supplements are available for figure 6:

**Figure supplement 1**. No increase in apical/basal astral microtubules of BPs upon wt LGN overexpression combined with minimal taxol incubation.

of symmetric vs asymmetric divisions with respect to the distribution of the apical domain, identified by the absence of cadherin staining (*Kosodo et al., 2004*). The previously published low percentage of asymmetric divisions was confirmed with BRO culture in controls (*Figure 9H* top bar, white = 12%). However, with 30 pM nocodazole, the proportion of asymmetric divisions in proliferating APs increased to a level similar to control neurogenic divisions (*Figure 9H*, second and third bar, white = 60% vs 62%). Nocodazole at 30 pM did not increase much the proportion of asymmetric divisions in neurogenic APs (*Figure 9H* bottom bar, white = 69%), showing that this nocodazole concentration mostly influences the division symmetry of proliferating APs. Together, these data show that 30 pM nocodazole is sufficient to induce mitotic proliferating APs to behave as neurogenic ones with respect to the number of apical/basal astrals (*Figure 6A–F*), spindle orientation variability (*Figure 9A–E*), and cell division symmetry (*Figure 9F–H*).

## Reducing the number of apical/basal astral microtubules increases basal progenitors, basal mitoses, and neurogenesis

An increase in asymmetric AP divisions could lead to changes in daughter cell fate, such as more AP daughter cells becoming BPs, and eventually to an increase in neurogenesis. To detect changes in the populations of progenitors and neurons, E13.5 BRO culture was used with or without 30 pM nocodazole. Compared to the DMSO control, incubation with 30 pM nocodazole for 24 hr resulted in a decrease in the proportion of APs, as identified by being in the ventricular zone (VZ) and having a characteristic transcription factor expression pattern (Pax6-positive, Tbr2-negative; *Figure 10A–C*, 62.5 ± 1.8% vs 53.8 ± 0.7%). Interestingly, a concomitant increase was observed in the proportion of Tbr2-positive newborn BPs in the VZ (*Figure 10A–C*, 34.3 ± 2.1% vs 43.1 ± 2.4%). These results indicate that more AP daughter cells became BPs upon an induced increase in asymmetric divisions. This is consistent with results showing an increase in BPs upon perturbation of other factors that also increase AP spindle orientation variability (*Fish et al., 2006*; *Konno et al., 2008*; *Postiglione et al., 2011*; *Shitamukai et al., 2011*).

To complement these data, we analysed the expression of the neurogenic marker Tis21::GFP in VZ progenitors (*Figure 10—figure supplement 1A–C*). Compared to controls, 30 pM nocodazole increased the proportion of Tis21::GFP-positive cells in the VZ (58.8 ± 5.1% vs 68.2 ± 2.4%). These cells may well comprise not only BP daughters but also AP daughters originating from Tis21::GFP-negative AP divisions that had become increasingly asymmetric. We conclude that minimal nocodazole-induced asymmetric AP divisions result in increased Tis21::GFP expression in daughter cells, consistent with these daughters being neurogenic progenitors.

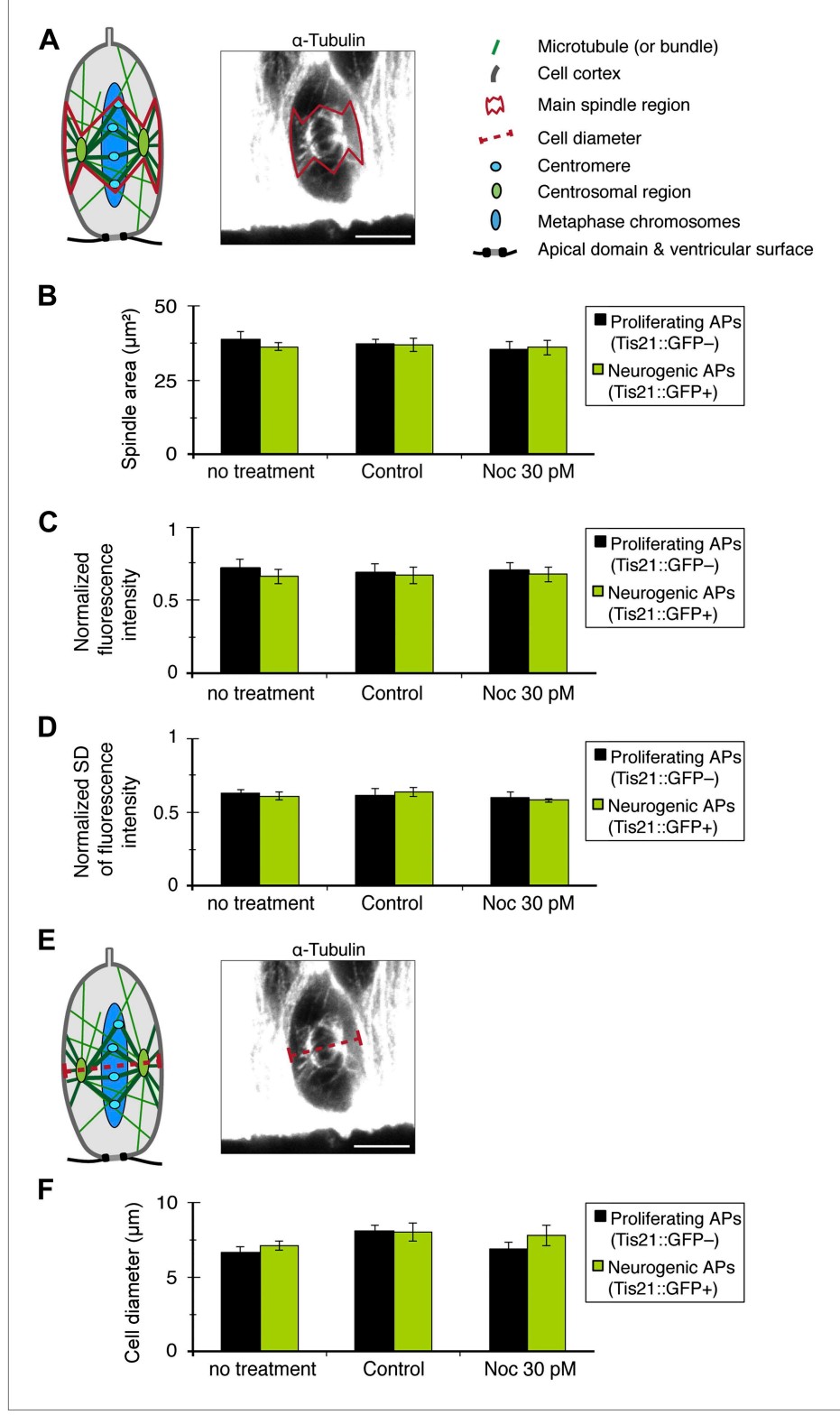

**Figure 7**. Key spindle and cell features in APs are not altered by 30 pM nocodazole. All measurements were performed on the same cell images from which the astral microtubule data were obtained (see *Figures 2 & 4*). (**A**) Cartoon and example image (α-tubulin immunofluorescence) of the mitotic spindle in an AP, indicating the main spindle region (solid red line) where α-tubulin immunofluorescence measurements were performed to control for *Figure 7. Continued on next page*

*Figure 7. Continued*

variations in spindle features (**B–D**), between proliferating and neurogenic APs, either without (no treatment) or with BRO culture of Tis21::GFP E14.5 forebrains, with either solvent (DMSO) only (Control) or with 30 pM nocoda-zole (Noc). The main spindle region was defined on the basis of the cell body shape and the centrosomal region (both revealed by α-tubulin immunofluorescence) and the apical- and basal-most (peri-)centromeric foci (DAPI staining) (see *Figure 2*A). The α-tubulin immunofluorescence used for illustration on the right side in **A** and **E** is the same as in *Figure 2*C. The key in **A** applies also for the cartoon in **E**. To obtain values from the central part of the spindle, the mean of the measurements from the 2 central-most 0.75 µm confocal sections was obtained. (**B**) Quantification of the area of the main spindle region. (**C**) Quantification of the amount of spindle microtubules, as obtained from the mean fluorescence intensity in the main spindle region normalized to the mean fluorescence intensity in the entire image. (**D**) Spatial distribution of α-tubulin in the main spindle region, as revealed by the mean standard deviation (SD) between the per-pixel fluorescence intensity values in the main spindle region, normalized to the mean fluorescence intensity of the main spindle region. (**E**) Cartoon and example image of the mitotic spindle showing the AP diameter measurement at the spindle plane (red dashed line). (**F**) AP cell body diameter at the spindle plane, indicating also spindle length. (**B–D, F**) Significant differences were not found with K–W with DMC post test. Error bars are SEM. Scale bars = 5 µm. See also *Figure 7—source data 1* and *Figure 7—figure supplement 1*.

The following source data and figure supplements are available for figure 7:

**Source data 1**. Table with the values for the graphs in *Figure 7*.

**Figure supplement 1**. Key subcellular, cellular, and tissue features in APs are not significantly altered by 30 pM nocodazole.

---

Of note, the increase in newborn BPs in the VZ did not yet result in a detectable increase in inter-phase Tbr2-positive or Pax6-positive BPs in the subventricular zone (SVZ) after 24 hr (Tbr2, control: 80.1 ± 2.0%, nocodazole: 82.3 ± 5.9%; Pax6, control: 31.9 ± 8.2%, nocodazole: 28.9 ± 12.7%), despite the higher number of mitoses in the SVZ (see below). This presumably reflects known differences in cell cycle length between progenitor subtypes (see 'Discussion').

A prediction from the increase in newborn BPs after minimal nocodazole is that the number of basal mitoses would increase. Confirming this, more mitoses were found in the SVZ with 30 pM noco-dazole than in controls (phosphorylated histone 3 (PH3) immunoreactivity; *Figure 10D–F*, 1.96 ± 0.16 vs 1.32 ± 0.029). This was corroborated with a second, independent mitosis marker (phosphovimentin (pVim); *Figure 10—figure supplement 1D–F*, 2.08 ± 0.25 vs 1.28 ± 0.05).

Importantly, the total number of mitoses did not change significantly with minimal nocodazole when analysed by either PH3 (*Figure 10F*, control: 7.98 ± 0.57, nocodazole: 8.22 ± 0.58) or by pVim (*Figure 10—figure supplement 1F*, control: 7.80 ± 0.52, nocodazole: 8.09 ± 0.42). In addition, 30 pM nocodazole did not result in more apoptosis (*Figure 10—figure supplement 1G–I* and 'Material and methods'). Also, all mitotic phases proceeded normally and at the same rate with 30 pM nocoda-zole as they did in controls (*Figure 9A–D*), and cellular morphology at mitosis was unperturbed (*Figure 6C,D*, *Figure 9F,G*). These data show that normal mitotic progression and the cell cycle were unaffected by 30 pM nocodazole. Importantly, the converse manipulation of microtubules by using 25 pM taxol did not affect mitotic progression (*Figure 8A–F*) or morphology (*Figure 6F,G,I,J*) either.

The increase in neurogenic progenitors, notably newborn Tbr2-positive BPs, predicted an increase in neuron generation. Counting the number of cells positive for the neuronal marker Tbr1 confirmed this. More neurons were found in the presence of 30 pM nocodazole than in the presence of only DMSO (*Figure 10G–I*, 381.2 ± 11.4 vs 327.2 ± 12.0). Together, these results show that an acute reduc-tion in the number of apical/basal astrals that leads to an increase in asymmetric AP divisions can increase BPs and neurogenesis.

## Discussion

Our study reports a novel concept regarding the composition and role of astral microtubule popula-tions in the symmetric vs asymmetric division of polarized stem and progenitor cells. Specifically, we use the established model of mouse neocortex APs to show that the transition between sym-metric and asymmetric neural stem cell division in cells with minor spindle variations is mediated by a

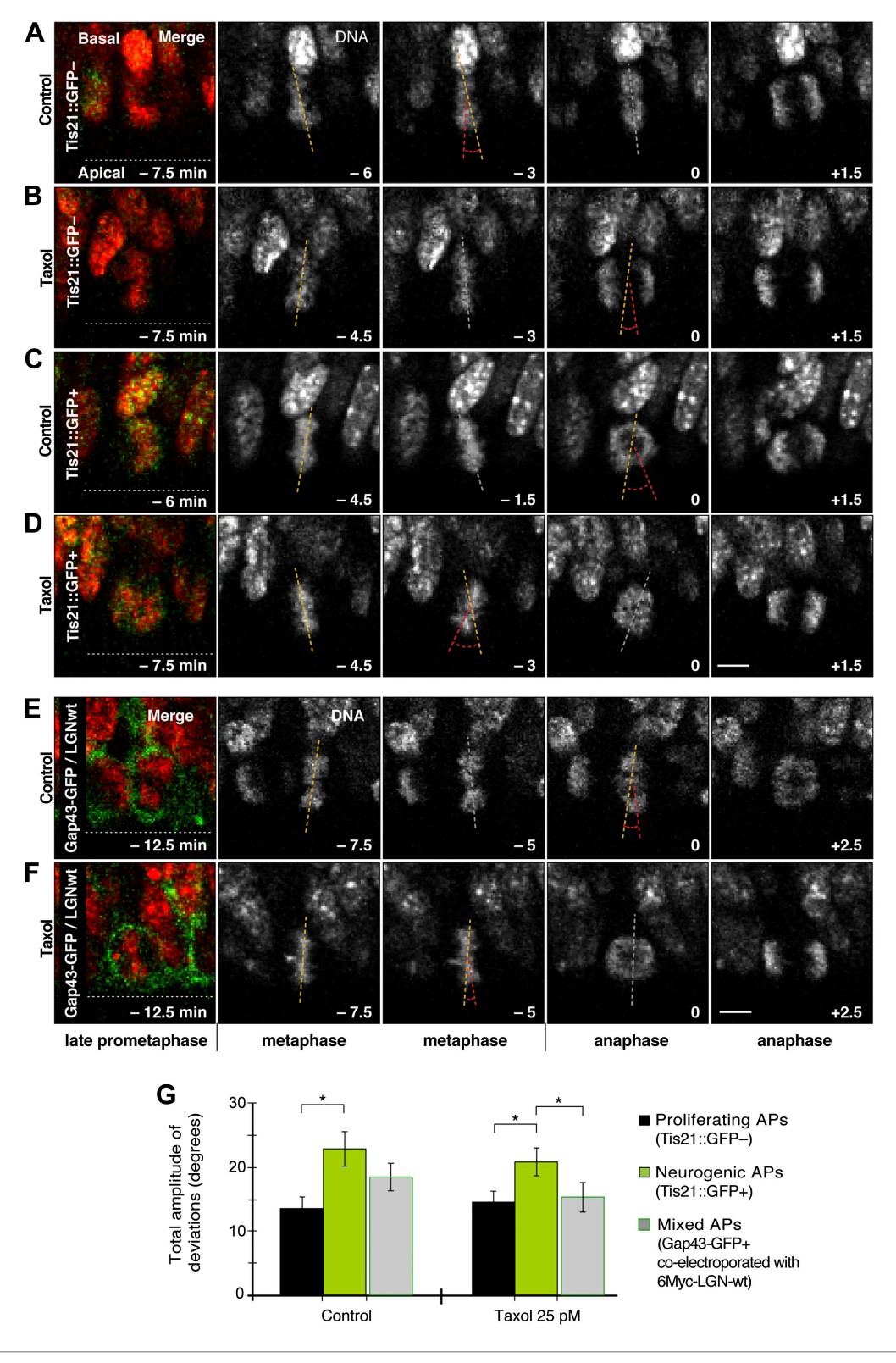

**Figure 8**. Dynamic spindle orientation variability decreases with minimal taxol incubation combined with LGN-wt overexpression, but not with taxol alone. **A–D**) E14.5 forebrains from Tis21::GFP mice were incubated on the microscope stage with either solvent (DMSO) only (Control) or with 25 pM taxol. Live tissue imaging of spindle
*Figure 8. Continued on next page*

*Figure 8. Continued*

orientation, as reported by chromosome plate orientation, in organotypic slice culture of coronal sections from Tis21::GFP mouse dorsolateral telencephalon. Time-lapse is 1.5 min. 0 min corresponds to anaphase onset. Apical progenitors underwent either proliferative (**A**, **B**, Tis21::GFP–) or neurogenic (**C**, **D**; Tis21::GFP+) divisions. Merge: chromosomes in red, EGFP in green. Chromosome plate orientation was determined by measuring angular deviations from the apico-basal axis (0°, corresponding to vertical), which runs perpendicular to the apical ventricular surface (90°, horizontal white dotted lines in **A**–**F**). Vertical or oblique dashed lines indicate chromosome plate orientations in metaphase or anaphase onset; maximal deviation angles for each plate were set by the orientation at an early time-point (yellow) and a later time-point (red), which are quantified in G. White dashed lines indicate intermediate orientations. Scale bar = 5 μm. (**E**, **F**) Similar to A,B and C,D, but with E13.5 wt forebrains subjected to in utero electroporation of 6Myc-LGN-wt together with GAP43-EGFP and 18 hr later (E14.25) incubated for 2.5 hr in organotypic slice culture with either DMSO only (control) or 25 pM taxol. Time-lapse is 2.5 min. (**G**) Mean ± SEM of the maximal amplitude of deviations for proliferating (Tis21::GFP–) vs neurogenic (Tis21::GFP+) APs or of APs electroporated with LGN-wt together with GAP43-EGFP, in control or 25 pM taxol incubations. *p<0.05, K–W with DMC post test, n = 20 progenitors per category, from two independent litters and experiments.

The following figure supplements are available for figure 8:

**Figure supplement 1**. No increase in dynamic spindle orientation variability of BPs upon wt LGN overexpression combined with minimal taxol incubation.

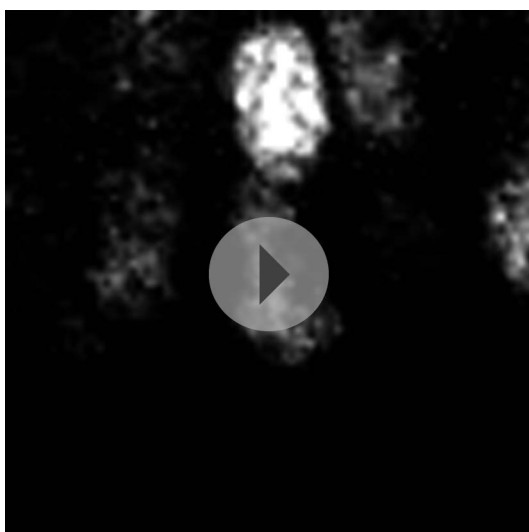

**Video 4**. Dynamic spindle orientation variability in a representative proliferating AP incubated with DMSO only at 1.5 min time resolution. Related to *Figure 8A*. Live tissue imaging of spindle orientation, as reported by the chromosome plate (DNA) orientation, in organotypic slice culture of dorsolateral telencephalon coronal sections, from an E14.5 Tis21::GFP mouse. Total time elapsed 15 min.

reduction in the number of astral microtubules that reach the cell cortex. Strikingly, this reduction in astrals is not homogeneous, but happens only in the astrals that reach the basal and apical cell cortex. This shows that a distinct subpopulation of astrals, the apical/basal astrals, governs spindle orientation.

Our selective reduction in the number of apical/basal astrals, by using minimal nocodazole, mimics the reduction observed in vivo and increases the proportion of asymmetric AP divisions and the pool of BPs, ultimately leading to more neurogenesis. To the best of our knowledge, this reduction is also the earliest change reported thus far in the cytoarchitecture of APs that switch from proliferation to neurogenesis. A homogeneous decrease in astrals, including those reaching the central cell cortex, happens only in BPs, which, accordingly, have the highest spindle orientation variability (*Figure 10J*). The spatial, temporal, and cell-type specific regulation of astrals is therefore key to maintain the population of proliferating APs for normal brain development.

A new model is therefore proposed to explain how spindle and cleavage plane variability increases during mammalian neurogenesis (*Chenn and McConnell, 1995*; *Haydar et al., 2003*; *Kosodo et al., 2004*; *Konno et al., 2008*). The subtle mammalian increase in variability is fundamentally different to the large and defined 90° re-orientation in *Drosophila* neuroblasts (*Knoblich, 2008*; *Siller and Doe, 2009*). We show that, as mammalian APs switch to neurogenesis, the spindle becomes less restricted to an orientation precisely perpendicular to the apico-basal axis (*Sauer, 1935*; *Smart, 1972a*, *1972b*; *Huttner and Brand, 1997*) as a result of fewer apical/basal astrals anchoring the spindle to the cell cortex (*Figure 10J*). The orientation of spindles that become less anchored is then more sensitive to intra- and extracellular forces that can induce tilts, including those resulting from the dense packing of dynamic cells in the developing cortical wall.

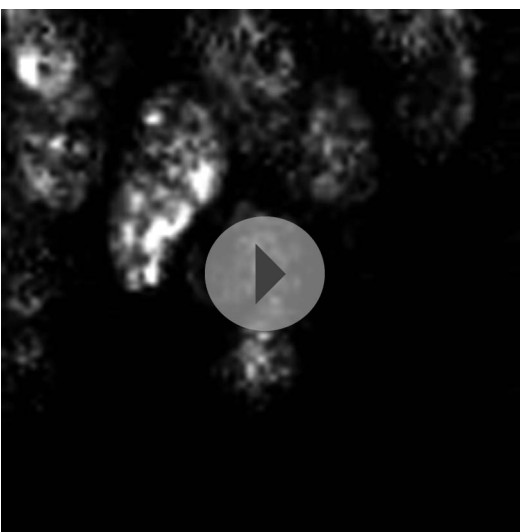

**Video 5**. Dynamic spindle orientation variability in a representative proliferating AP incubated with 25 pM taxol at 1.5 min time resolution. Related to **Figure 8B**. Live tissue imaging of spindle orientation, as reported by the chromosome plate (DNA) orientation, in organotypic slice culture of dorsolateral telencephalon coronal sections, from an E14.5 Tis21::GFP mouse. Total time elapsed 13.5 min.

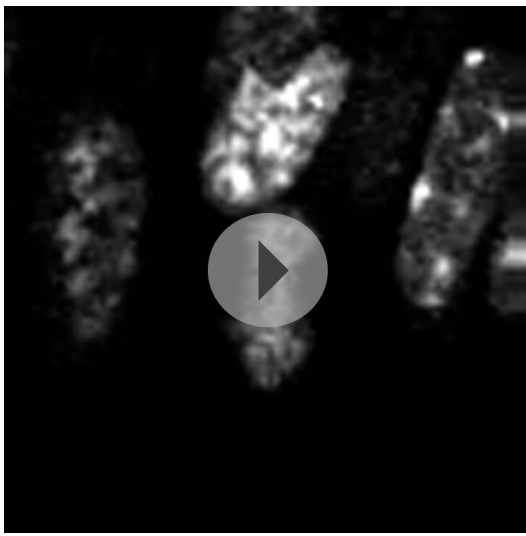

**Video 6**. Dynamic spindle orientation variability in a representative neurogenic AP incubated with DMSO only at 1.5 min time resolution. Related to **Figure 8C**. Live tissue imaging of spindle orientation, as reported by the chromosome plate (DNA) orientation, in organotypic slice culture of dorsolateral telencephalon coronal sections, from an E14.5 Tis21::GFP mouse. Total time elapsed 10.5 min.

The increase in BPs upon increasing spindle orientation variability by minimal nocodazole treatment is consistent with previous studies showing more BPs in mice where spindles were tilted by perturbing the LGN binding partner Inscuteable (*Postiglione et al., 2011*), or LGN itself (*Konno et al., 2008*; *Shitamukai et al., 2011*). The increase in mitotic BPs observed upon LGN perturbation was associated with an increasing appearance of Pax6-positive and of Tbr2-positive cells in the SVZ (*Konno et al., 2008*; *Shitamukai et al., 2011*). Inscuteable overexpression increased the number of mitotic, Tbr2-positive cells in the SVZ (*Postiglione et al., 2011*). The reduction in apical/basal astrals shown here generated more newborn Tbr2-positive BPs in the VZ and mitotic BPs in the SVZ. Thus, these three distinct approaches that tilt the spindle to increasingly deviate from an orientation perpendicular to the apico-basal AP axis yield similar (albeit not identical) phenotypes, that is, an increase in delaminated progenitors that exhibit BP features and undergo basal mitosis. The more specific differences could reflect the distinct types of manipulation, i.e., the genetic ablation of LGN vs Inscuteable overexpression vs pharmacological perturbation of apico/basal astrals. Taken together, our data decisively support the concept that a small change in cleavage plane orientation of APs can be sufficient to cause a fate change in their progeny (*Fish et al., 2006*; *Postiglione et al., 2011*).

Importantly, the delaminated progenitors increasingly observed upon minimal nocodazole treatment showed Tbr2 immunoreactivity and underwent basal mitosis, hence showing a number of features that are typical of BPs rather than APs. Nevertheless, these newborn Tis21::GFP-positive BPs, generated by just tilting the spindle, may still temporarily exhibit certain AP features, such as an AP-like cell cycle length, which is shorter than that of mature BPs (*Arai et al., 2011*). These newborn BPs may therefore undergo mitosis soon after arrival in the SVZ, which would explain its increased mitoses without a significant increase in Tbr2-positive interphase cells. A shorter than normal cell cycle of the BPs generated by minimal nocodazole would also be consistent with the newborn neurons increasingly observed after 24 hr of this treatment being derived not only directly from the asymmetrically dividing APs, but also indirectly from these BPs.

Spindle orientation studies have faced the challenge of establishing whether the phenotypes observed could be solely attributed to changes in orientation, or to other effects on the spindle and other cellular components. This study aimed at specifically perturbing spindle orientation while avoiding other effects. For this, a minimal concentration

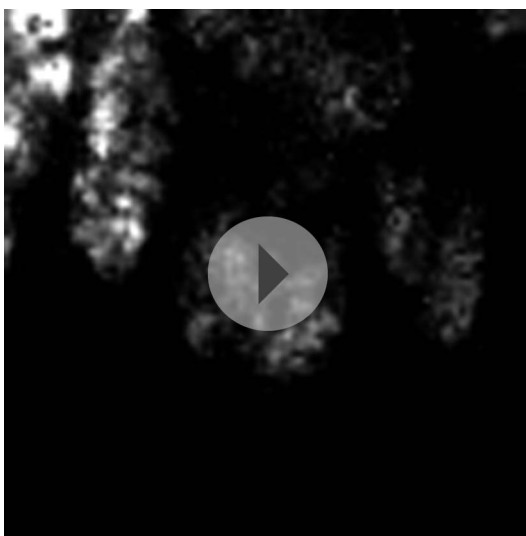

**Video 7**. Dynamic spindle orientation variability in a representative neurogenic AP incubated with 25 pM taxol at 1.5 min time resolution. Related to *Figure 8D*. Live tissue imaging of spindle orientation, as reported by the chromosome plate (DNA) orientation, in organotypic slice culture of dorsolateral telencephalon coronal sections, from an E14.5 Tis21::GFP mouse. Total time elapsed 12 min.

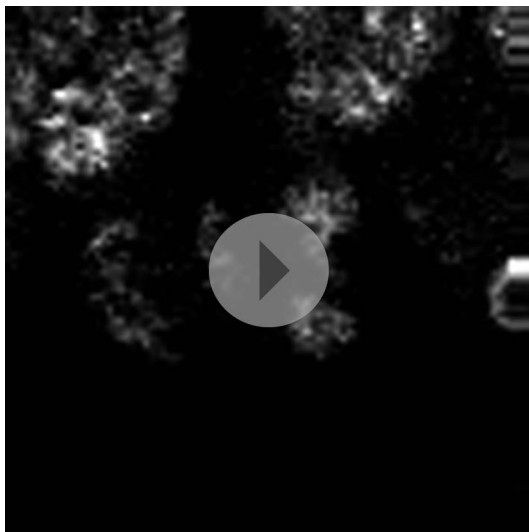

**Video 8**. Dynamic spindle orientation variability in a representative AP incubated with DMSO only and co-electroporated with GAP43-EGFP and 6Myc-LGN-wt. Related to *Figure 8E*. Live tissue imaging of spindle orientation, as reported by the chromosome plate (DNA) orientation, in organotypic slice culture of dorsolateral telencephalon coronal sections, from an E14.5 wt mouse. Time-lapse 3 min. Total time elapsed 17.5 min.

of the highly specific microtubule polymerization inhibitor nocodazole was identified that was several orders of magnitude lower than previously used in spindle studies. This minimal nocodazole decreased only the number of apical/basal astrals, those detected to change when APs switch from proliferation to neurogenesis, but did not affect other types of microtubules, notably the central astrals. This suggests that these astrals are more sensitive to depolymerisation, perhaps due to being highly dynamic. Moreover, this decrease was titrated to the physiological level observed in unperturbed neurogenic APs, and other effects, notably on progenitor and spindle features, mitotic morphology, and progression, were not detected. Consistent with this, a similarly low concentration of the specific microtubule stabilizer taxol showed the opposite effect, increasing the number of apical/basal astrals. Therefore, treatment with minimal concentrations of compounds acting specifically on microtubules constitutes an alternative approach to genetic manipulations, allowing direct probing of spindle orientation per se and of the symmetry of cell division in tissue.

Furthermore, LGN is here shown to regulate the spatial and temporal abundance of astrals that reach the cell cortex, and thereby the transition between proliferation and neurogenesis. Mammalian LGN has been shown to participate in the anchoring of the spindle to the cell cortex in cultured MDCK cells, as part of a complex with NuMA and dynein, by mediating the attachment of astrals to cell cortical Gαi subunits (*Du and Macara, 2004*). Consistent with this role, previous studies showed that LGN is important for normal cleavage plane and symmetric AP divisions in the mouse (*Konno et al., 2008*; *Shitamukai et al., 2011*) and the chick (*Morin et al., 2007*; *Peyre et al., 2011*), but how LGN achieved this was unknown. Using an LGN KO mouse (*Konno et al., 2008*), LGN is now shown to be necessary for APs to have a normal number of apical/basal astrals. Importantly, lack of LGN did not affect the number of central astrals, corroborating that loss of apical/basal astrals is sufficient to increase spindle deviations and asymmetric divisions. These data were complemented by overexpressing a dominant-negative version of LGN that interacts with cell cortical Gαi, but cannot interact with NuMA and dynein to anchor microtubules (*Morin et al., 2007*; *Shitamukai et al., 2011*). This also reduced apical/basal astral abundance. Conversely, when wt LGN was overexpressed in the presence of the microtubule-stabilizing compound taxol, neurogenic APs could be switched back to a mitosis behaviour characteristic of proliferating APs. Together, these data show that LGN is required for

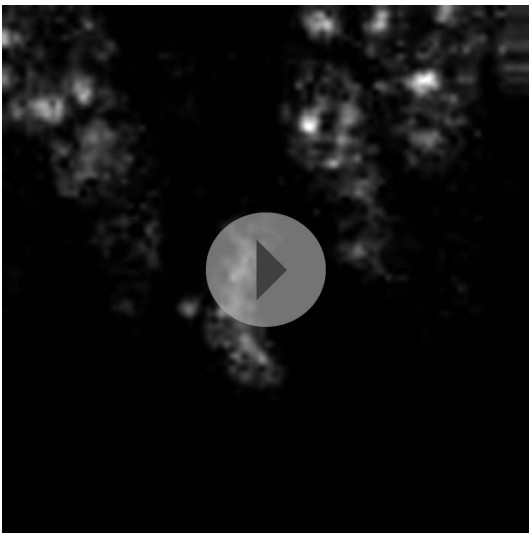

**Video 9**. Dynamic spindle orientation variability in a representative AP incubated with 25 pM taxol and co-electroporated with GAP43-EGFP and 6Myc-LGN-wt. Related to *Figure 8F*. Live tissue imaging of spindle orientation, as reported by the chromosome plate (DNA) orientation, in organotypic slice culture of dorsolateral telencephalon coronal sections, from an E14.5 wt mouse. Time-lapse 3 min. Total time elapsed 20.5 min.

maintaining symmetric AP divisions through cortical anchoring of apical/basal astrals.

Our data also show how different cortical localizations of LGN could regulate the abundance of astrals that reach different regions of the cell cortex. When APs switched to neurogenesis, the enrichment of LGN at the basal cell cortex was largely reduced, and it was further reduced all around the cell cortex in BPs. This closely corresponds to the abundance pattern of astrals and also to the levels of spindle orientation variability in these different progenitors. A spatially selective loss of cortical LGN could therefore reduce cortical anchor points and lead to fewer astrals in specific regions of the cell cortex (*Figure 10J*).

The present study showing the crucial role of a specific subset of LGN-dependent astrals in the control of spindle and cleavage plane orientation also builds upon previous studies on another microtubule-interacting protein, the lissencephaly protein Lis1 (*Sapir et al., 1999*; *Faulkner et al., 2000*; *Tsai et al., 2005*; *Yingling et al., 2008*; *Bi et al., 2009*). In the neural tube of mice without Lis1, cleavage planes of APs were more tilted (*Yingling et al., 2008*). Also, cultured embryonic fibroblasts lacking Lis1 showed general defects in interphase and mitotic microtubules, including astrals (*Yingling et al., 2008*). The present findings suggest that the altered cleavage plane orientation in Lis1-deficient APs could be due, at least in part, to a loss of apical/basal astrals.

If apical/basal astral microtubules are key to controlling spindle and cleavage orientation, what then is the role of the central astrals unaffected by the proliferation-to-neurogenesis transition of progenitors (*Figure 10J*) and that remained constant upon minimal nocodazole treatment or functional LGN ablation? An intriguing scenario is that this baseline number of remaining astrals is required for the primary establishment and structural stability of a spindle. If so, our data suggest a way how cells differentially regulate core aspects of spindle function vs those that are cell type-specific, notably spindle orientation.

## Materials and methods

### Mice

All observations were performed in the dorsolateral telencephalon of mouse embryos, at a medial position along the rostro-caudal axis. All mice, wt or mutant, were C57BL/6. Animals were kept pathogen-free at the Biomedical Services Facility (BMS) of the MPI-CBG. All experiments were performed according to the German Animal Welfare Legislation. Embryonic day (E) 0.5 was set at noon on the day of vaginal plug identification. Neurogenic progenitors were identified in knock-in mice heterozygous for EGFP expressed under the control of the Tis21 promoter (*Haubensak et al., 2004*). To examine astral microtubules (astrals) in mitotic cells without LGN, LGN knock-out (KO) mice (*Konno et al., 2008*) were used.

### BRO culture

A brain culture method was developed that is termed BRO (**B**rain **R**otation **O**rganotypic) culture, which includes aspects of previous culture systems (*Nomura and Osumi, 2004*; *Attardo et al., 2008*; *Schenk et al., 2009*) and is as follows: Brains of E13.5–E14.5 embryos were dissected and placed in Tyrode solution (Sigma T2145, Germany) at 37°C, where most of the meninges were surgically removed. Whole forebrains were dissected and transferred to 20 ml glass flasks with 1 ml brain culture medium (see 'Live

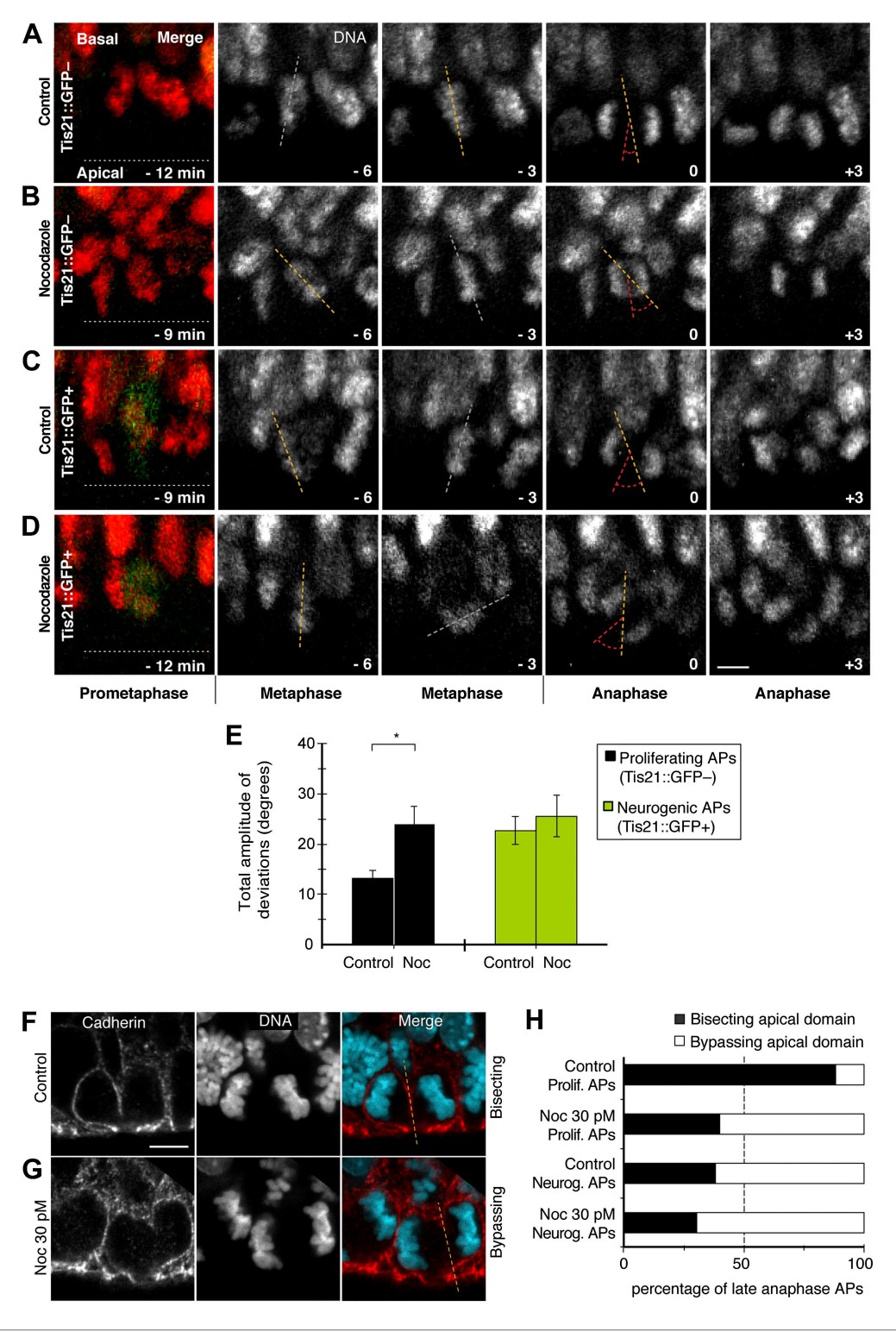

**Figure 9**. Dynamic spindle orientation variability and asymmetric AP divisions increase with minimal nocodazole. E14.5 forebrains from Tis21::GFP mice were incubated on the microscope stage with either solvent (DMSO) only (Control) or with 30 pM nocodazole (Noc). (**A–D**) Live tissue imaging of spindle orientation, as reported by chromosome plate orientation, in organotypic slice culture of coronal sections from E14.5 Tis21::GFP mouse dorsolateral telencephalon. 0 min corresponds to anaphase onset. Apical progenitors underwent either

*Figure 9. Continued on next page*

*Figure 9. Continued*

proliferative (**A**, **B**, Tis21::GFP–) or neurogenic (**C**, **D**; Tis21::GFP+) divisions. Merge: chromosomes in red, EGFP in green. Chromosome plate orientation was determined by measuring angular deviations from the apico-basal axis (0°, corresponding to vertical), which runs perpendicular to the apical ventricular surface (90°, horizontal white dotted lines in **A**–**D**). Time-lapse is 3 min. Vertical or oblique dashed lines indicate chromosome plate orientations in metaphase or anaphase onset; maximal deviation angles for each plate were set by the orientation at an early time-point (yellow) and a later time-point (red), which are quantified in E. White dashed lines indicate intermediate orientations. Scale bar = 5 µm. (**E**) Mean ± SEM of the maximal amplitude of deviations for proliferating (Tis21::GFP–) vs neurogenic (Tis21::GFP+) APs, either in control or nocodazole incubations, *p<0.05, K–W with DMC post test, n = 20 progenitors per category, from 3 independent litters and experiments. See also *Figure 7*. (**F**, **G**) Cadherin immunofluorescence combined with DNA staining (DAPI) to reveal symmetric vs asymmetric divisions of anaphase APs with regard to the apical domain, as reported by bisecting or bypassing, respectively, of the cadherin-negative apical plasma membrane (dashed lines). Merge: cadherin, red; DNA, cyan. (**H**) Percentages of late anaphase APs, with advanced cytokinetic cleavage furrow, prospectively bisecting or bypassing their apical domain; n = 20 progenitors per category, from three independent litters and experiments. Scale bars = 5 µm.

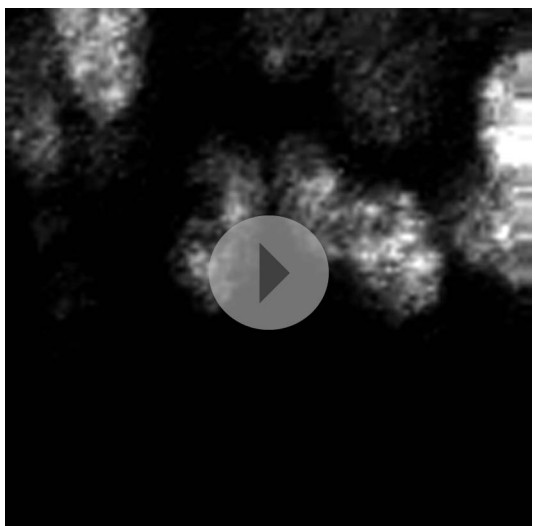

**Video 10**. Dynamic spindle orientation variability in a representative proliferating AP incubated with DMSO only. Related to *Figure 9A*. Live tissue imaging of spindle orientation, as reported by the chromosome plate (DNA) orientation, in organotypic slice culture of dorsolateral telencephalon coronal sections, from an E14.5 Tis21::GFP mouse. Time-lapse 3 min. Total time elapsed 15 min.

Tissue Imaging'). Flasks were connected to a whole-embryo culture incubator (RKI Ikemoto, Japan) and maintained at 37°C with an atmosphere of 60% $O_2$, 5% $CO_2$, 35% $N_2$, and at 30 rpm, allowing tissue rotation and nutrition without mechanical damage. After 1 hr of equilibration, pre-warmed brain culture medium containing 10 nM nocodazole or taxol and 1% DMSO, added from a 1 µm stock solution in DMSO, or brain culture medium containing 1% DMSO only (control), was added to the culture in amounts appropriate to obtain the indicated final concentration. To count astrals, E14.5 forebrains were incubated for 2.5 hr. To analyse differences in the populations of progenitors, mitotic cells, and neurons, E13.5 forebrains were incubated for 24 hr, during which the tissue remained intact without a significant cell death increase (*Figure 10—figure supplement 1*). Brains were then fixed in 4% PFA in phosphate buffer for 2 hr at room temperature followed by 4°C overnight.

## In utero electroporation and plasmids

In utero electroporation (*Takahashi et al., 2002*; *Fish et al., 2006*) was used to introduce the dominant-negative LGN-C or LGN-wt (*Shitamukai et al., 2011*) into APs. E13.5 pregnant mice were fully anesthetized with isofluorane and received a subcutaneous analgesic injection. The uterus was then exposed by surgically opening the peritoneal cavity, maintaining moisture and 37°C temperature. Embryos were injected intraventricularly with a 0.1% solution of fast green (Sigma) in sterile PBS, containing 0.2–0.5 µg/µl of LGN plasmid, plus mCherry or GAP43-EGFP, driven by the CAGGS constitutive promoter (*Niwa et al., 1991*) to locate electroporated regions and cells. Electroporations were with six 50 ms pulses of 30 mV at 1 s intervals. Detection of LGN and the fluorescent reporter occupied two of the four available fluorescence channels, with DAPI and tubulin occupying the other two, so these experiments could only be performed with wt and not with Tis21::GFP mice. For consistency, this scheme was followed for the live imaging of organotypic slice cultures (see 'Live tissue imaging').

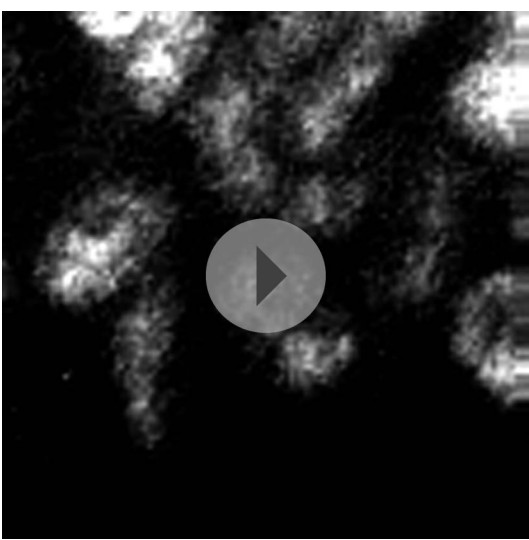

**Video 11**. Dynamic spindle orientation variability in a representative proliferating AP incubated with DMSO only. Related to **Figure 9B**. Live tissue imaging of spindle orientation, as reported by the chromosome plate (DNA) orientation, in organotypic slice culture of dorsolateral telencephalon coronal sections, from an E14.5 Tis21::GFP mouse. Time-lapse 3 min. Total time elapsed 12 min.

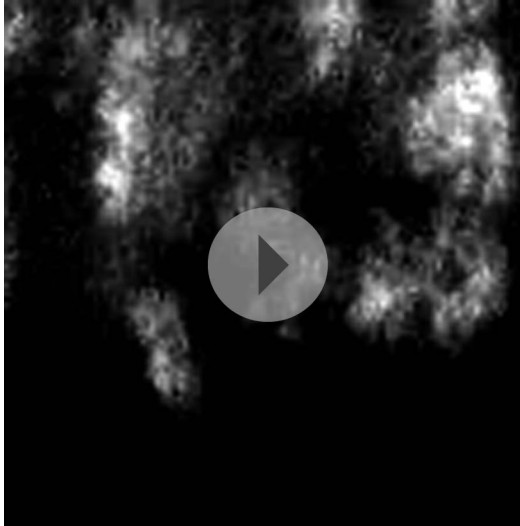

**Video 12**. Dynamic spindle orientation variability in a representative neurogenic AP incubated with 30 pM nocodazole. Related to **Figure 9C**. Live tissue imaging of spindle orientation, as reported by the chromosome plate (DNA) orientation, in organotypic slice culture of dorsolateral telencephalon coronal sections, from an E14.5 Tis21::GFP mouse. Time-lapse 3 min. Total time elapsed 12 min.

## Microscopy

Images were acquired with a Zeiss LSM 510 Duoscan laser-scanning confocal microscope or a Zeiss LSM 780 NLO 2-photon laser-scanning microscope with an automated tuning pulsed Ti:S Chameleon Vision II laser (Coherent), using 63× Plan-Apochromat 1.4 N.A. oil or 40× C-Apochromat 1.2 N.A. W objectives (Carl Zeiss, Germany). Multiposition and multidimensional imaging was controlled as described (**Rabut and Ellenberg, 2004**).

### Live tissue imaging

Freshly dissected E14.5 forebrains were embedded in 3% low melting-point agarose (Sigma) in PBS. Coronal sections (200 µm) were obtained by vibratome (Leica, Germany) sectioning and embedded in type Ia collagen (Cellmatrix, Japan) diluted to 1.7 mg/ml with DMEM and buffer according to the manufacturer's instructions. Embedded sections were mounted in the 14 mm glass bottom microwell of 35 mm dishes (MatTek, Germany) and incubated for 45 min at 37°C for collagen solidification. Sections in the dish were further cultured for observation on a microscope stage incubation chamber (Pecon, Germany) at 37°C in 1 ml of brain culture medium, composed of Neurobasal medium (Invitrogen, San Francisco, CA) supplemented with 10% immediately centrifuged rat serum (Charles River Labs, Japan), 1× N2 supplement (Invitrogen), 1× B27 supplement (Invitrogen), and 100 U/ml penicillin/streptomycin. The objective lens was kept at 37°C with an objective heater (Bioptechs, Butler, PA). Culture atmosphere was maintained at 40% $O_2$, 5% $CO_2$, 55% $N_2$. The incubation dish was covered with a semi-permeable membrane (H Sauer, Germany) and the edges sealed with silicone paste (GE Bayer, Germany), which allowed gas exchange while preventing evaporation. Stacks of typically 512 × 512 pixels × 3-6 optical sections (xyzt sampling: 0.28 × 0.28 × 2.5 µm × 1.5–5 min) were acquired for 4–10 hr, with Hoechst 33342 (Sigma) as vital DNA dye. For perturbation of astrals, nocodazole or taxol were added to the brain culture medium as described under BRO culture, 2 hr before the start of image acquisition. Potential phototoxicity effects were stringently controlled as described (**Mora-Bermudez and Ellenberg, 2007**).

### Immunofluorescence

The forebrains of mice at typically E14.5, but also E11.5, E13.5, and E16.5 were dissected and fixed in 4% PFA in phosphate buffer for 2 hr at room temperature followed by 4°C overnight. Pax6, Tbr2, Tbr1, PH3, pVim, and caspase 3 staining on

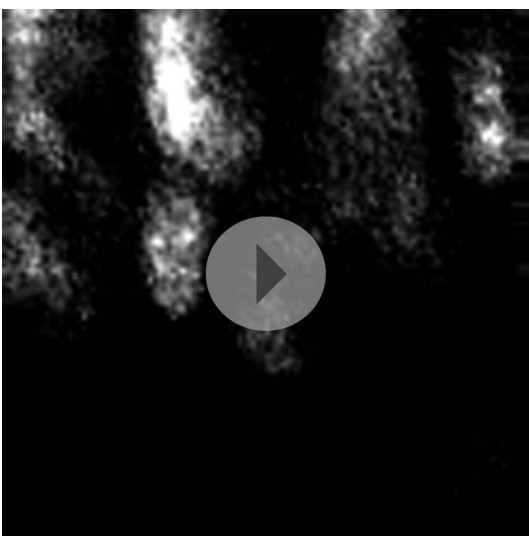

**Video 13**. Dynamic spindle orientation variability in a representative neurogenic AP incubated with 30 pM nocodazole. Related to *Figure 9D*. Live tissue imaging of spindle orientation, as reported by the chromosome plate (DNA) orientation, in organotypic slice culture of dorsolateral telencephalon coronal sections, from an E14.5 Tis21::GFP mouse. Time-lapse 3 min. Total time elapsed 15 min.

cryosections was performed as described (*Arai et al., 2011*); LGN staining was similar, except that forebrains were fixed with 0.5% PFA in phosphate buffer for 30 min at room temperature. For vibratome sections (GFP, Myc, α- & γ-tubulin, Arl13b, and cadherin stainings), fixed forebrains were embedded in 3% low melting-point agarose (Sigma) and 50 μm coronal sections were obtained, permeabilized with 0.1–0.3% Triton X-100 in PBS for 1 hr, quenched with 0.2 M glycine in PBS for 30 min, followed by blocking, primary, and secondary antibody incubations, and washings in PBS containing in addition 0.2% gelatin, 150 mM NaCl and 0.1–0.3% Triton X-100. Primary antibodies were incubated overnight at 4°C and secondaries for 2 hr at room temperature. Sections were mounted in Mowiol on glass slides. Stacks of typically 512 × 512 pixels × 15–20 optical sections (xyz sampling: 0.09 × 0.09 × 0.75 μm) were acquired for microtubule analysis, or 2048 × 2048 pixels × 5 optical sections (xyz sampling: 0.16 × 0.16 × 1.0 μm) for analysing cell populations in tissue.

## Antibodies

Mouse: mAb anti α-tubulin, (Sigma), mAb anti pan-cadherin (Sigma), mAb anti cleaved caspase 3 (Sigma), mAb anti phosphovimentin (Abcam, San Francisco, CA), mAb anti-Myc-A488 or -A555 (Millipore, Germany); rabbit: pAb anti Pax6 (Covance); pAb anti Tbr2 (Abcam), pAb anti Tbr1 (Millipore), pAb anti γ-tubulin (Sigma), pAb anti Arl13b (Proteintech, UK), pAb anti phosphorylated histone H3-S10 (PH3, Millipore), pAb anti LGN (*Konno et al., 2008*); goat: antibody anti EGFP (MPI-CBG). Secondary antibodies: Alexa Fluor 488 donkey anti goat IgG, Alexa Fluor 555 donkey anti rabbit IgG, Alexa Fluor 647 donkey anti rabbit IgG, and Alexa Fluor 647 donkey anti mouse IgG, all from Invitrogen. Staining: DAPI (Sigma).

## Image analysis

Images were analysed and prepared with ImageJ (http://imagej.nih.gov/ij/) and AIM software (Carl Zeiss). The brightness and contrast of images were recorded and adjusted linearly. To analyse spindle and cleavage orientation dynamics by live tissue imaging, a measurement was made for each time point of the angle between the main axis of the chromosome plates, typically along the apico-basal axis of the tissue, and the plane of the local apical ventricular surface. At anaphase onset, this chromosome plate angle is also practically identical to the angle of cleavage of the cell soma, and therefore directly reports on the symmetry or asymmetry of the cell division with respect to cellular features. The angles are given as deviations from full orthogonality with the local apical surface, as seen from a coronal perspective (*Figure 1A–F*, *Figure 10J*). The accuracy of angular measurements was corroborated along the 3D stack of optical sections. To quantitate the range of variability that spindle orientation can have in different progenitor types, the mean of the total angular amplitude of deviations from metaphase to anaphase onset was calculated (*Figure 1G*).

To quantitate variations in the cell cortical distribution of LGN, a contour of per-pixel fluorescence intensity values was obtained along the cell cortex, in immunofluorescence images of a single central optical section per 3D stack (*Figure 4K*). The start was at the centre of the apical region and continued clockwise around the cell soma. The cortical intensity values were obtained by subtracting the diffuse cytoplasmic signal from the intensities measured along the one pixel-wide contour. To compare equivalent values between different cells, 100 equidistant cortical intensity values along the normalized cell soma contours were calculated by linear interpolation. The area, fluorescence intensity, and standard deviation measurements of the spindle in *Figure 7* (see also *Figure 7—source data 1*) were performed as described (*Mora-Bermudez et al., 2007*).

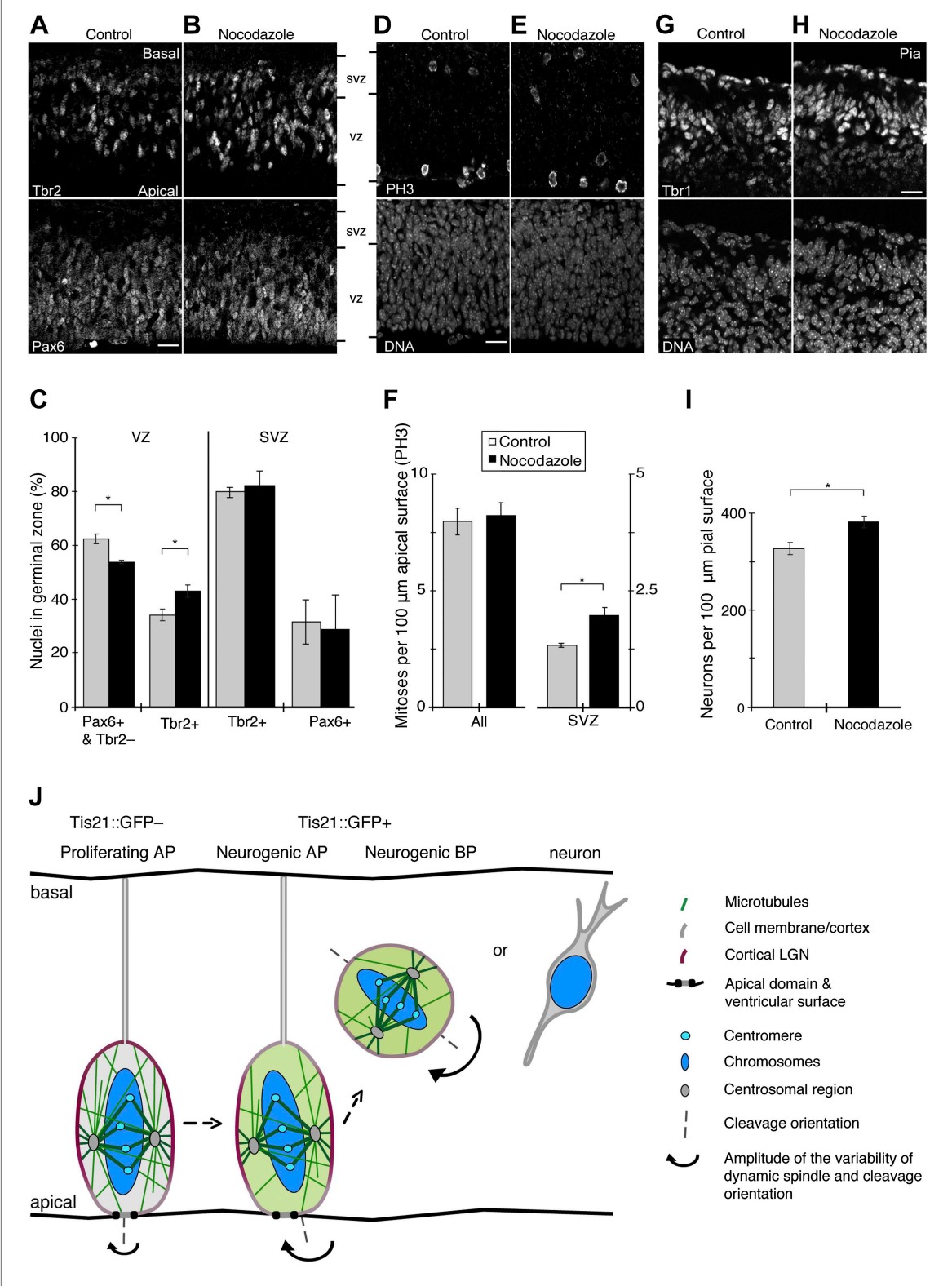

**Figure 10**. Increase in basal progenitors, basal mitoses, and neurogenesis with minimal nocodazole. E13.5 forebrains from wt mice were incubated for 24 hr in BRO culture, either with solvent (DMSO) only (Control) or with 30 pM nocodazole, followed by immunofluorescence of coronal sections of the dorsolateral telencephalon, acquisition of 1 μm confocal sections (**A, B, D, E, G, H**) and quantification (**C, F, I**). (**A, B**) Tbr2 (top) and Pax6 (bottom)

*Figure 10. Continued on next page*

*Figure 10. Continued*

double staining. VZ, ventricular zone; SVZ, subventricular zone. (**C**) Left: percentages of nuclei, identified by DAPI staining (not shown), in the VZ that are either resident APs (Pax6-positive and Tbr2-negative) or newborn BPs (Tbr2-positive). Control vs nocodazole: APs, *p=0.002; BPs, *p=0.004. Right: percentages of nuclei in the SVZ that are positive for Tbr2 or Pax6. (**D–E**) Staining for the mitosis marker PH3 (top) and DNA (DAPI, bottom). (**F**) Number and location of mitotic cells per 100 µm of apical (ventricular) surface; All, sum of mitoses in VZ and SVZ. Control vs nocodazole: SVZ, *p=0.031. (**G, H**) Staining for the neuronal marker Tbr1 (top) and DNA (DAPI, bottom). Note that the Tbr1 immunoreactivity basal to the cortical plate has been reported before (***Englund et al., 2005***); it may also reflect tissue stretching during cryosectioning. (**I**) Number of neurons per 100 µm of pial surface. Control vs nocodazole: *p=0.015. (**C, F, I**) One-tail *t*-tests, n = 3 independent litters and experiments; error bars are SEM. (**A, B, D, E, G, H**) Scale bars = 20 µm. See also ***Figure 10—figure supplement 1***. (**J**) Conceptual model of mitotic spindle orientation control by local astral microtubule abundance, in proliferating vs neurogenic progenitors. We propose that mitotic astral microtubules that reach the apical or basal cell cortex (light green rods) of neural stem and progenitor cells regulate the dynamics of spindle orientation variability. Like guy ropes do for a camping tent, these astrals help anchor the spindle to the cell cortex, most likely through interactions with factors that have specific cortical enrichments, such as LGN (red; colour intensity indicates local abundance in the three progenitor types shown). In APs undergoing proliferative division (Tis21::GFP–), these apical/basal astrals at relatively high abundance minimize deviations in spindle orientation, thereby maintaining the canonical cleavage orientation, perpendicular to the apical (ventricular) surface, and thus favouring symmetric proliferative divisions. In neurogenic (Tis21::GFP+) APs, and even more so in BPs, a reduction in the number of these astrals decreases cortical anchoring and increases random deviations in spindle orientation. For APs, this favours asymmetric neurogenic divisions that can generate a BP or a neuron. By contrast, astrals that reach the central cell cortex (medium green rods) of neural progenitors may be more involved in the fundamental establishment of a centrally located and functional bipolar spindle that can congress, and then faithfully segregate, chromosomes. (Dark green rods are kinetochore microtubules; for simplicity, other microtubule populations, such as interpolar microtubules, are omitted.)

The following figure supplements are available for figure 10:

**Figure supplement 1**. BRO culture with 30 pM nocodazole increases Tis21-positive VZ cells and basal mitoses, but not apoptosis.

## Quantitation of distinct populations of astral microtubules

To distinguish and count populations of astrals according to their spatial distribution, each set of confocal sections of a metaphase cell soma was divided into three regions: an apical, a central, and a basal region. The apical region was defined from the apical surface lining the ventricle to the parallel plane just before the first centromeric region on the metaphase plate (below pink dashed line in ***Figure 2A***). The central region that followed was defined to contain all centromeric regions, as revealed by the punctate DAPI staining of the centromeric and pericentromeric heterochromatin foci (between pink and blue dashed lines in ***Figure 2A***). The basal region extended after the last centromeric region until the basal end of the cell soma (above blue dashed line in ***Figure 2A***).

Comprehensive stacks of high-resolution serial confocal sections (see Microscopy) were examined from each progenitor in metaphase. All detectable astrals microtubules that emanated from the 2 centrosomes and reached the cell periphery were counted in each of the above-defined regions. These presumably are single microtubules, although bundles of microtubules cannot be excluded.

The apico-basal plane of the chromosome plates in APs can tilt with respect to the apical–basal axis of the tissue. However, it can also rotate around that axis, as seen from an 'en face' perspective (***Adams, 1996***; ***Haydar et al., 2003***). Therefore, in coronal sections, it is possible to observe spindles in a 'front' view, where all or most of the length of the spindle is visible in one section, or in a 'side' view, showing only one side of the spindle (***Figure 2A***). Spindles were therefore classified accordingly in these two, cell-biologically equivalent, perspective categories, a 'Front' view, when the main axis of the spindle was mostly parallel to the sectioning plane and fully visible within 1–3 confocal sections, and a 'Side' view, when the main axis of the spindle extended mostly along the sectioning axis (***Figure 2A***).

Astrals in the apical and basal regions could be reliably counted in cells with any such perspective. The central region of the spindle showed a dense α-tubulin staining, however, which in a 'Front' view made the distinction of many of its astrals very challenging. Therefore, astrals in the central region were counted in cells with a 'Side' view (***Figure 2A***). This may have led to an underestimation of the total number of central astrals, especially short ones oriented along the optical axis, albeit equally in all progenitor types.

In most mouse BPs, canonical apical cell polarity markers are lost (***Attardo et al., 2008***) and cell cortex regions can no longer be reliably defined as 'apical' or 'basal'. BPs were included in our analysis for comparative purposes, with the note that astrals that reach the region of the cell cortex facing mostly the ventricle are considered 'apically oriented', those that reached the region of the cortex facing mostly the pia 'basally oriented', and those that reach the region in between the previous regions 'centrally oriented' (***Figure 2H,I***; ***Figure 4G***).

## Data analysis

Datasets were tabulated and analysed using Excel (Microsoft, Redmond, WA) and GraphPad Prism (La Jolla, CA). Statistical tests: for two groups of observations, the Student's $t$-test for parametric analysis or the Mann–Whitney U-test for non-parametric analysis. For three or more groups, the one-way Analysis of Variance (ANOVA) with Tukey's Multiple Comparison (TMC) Test for parametric analysis, or the Kruskal–Wallis ANOVA (K–W) with Dunn's Multiple Comparison (DMC) Test for non-parametric analysis. Results were interpreted as statistically significant when $p < 0.05$.

## Selective perturbation of astral microtubules with minimal nocodazole

Nocodazole is a highly specific inhibitor of microtubule polymerization, and 30 pM is 2–3 orders of magnitude below the concentrations typically considered 'low', where the earliest signs of microtubule and cellular perturbations have been reported to appear (5–10 nM) (*De Brabander et al., 1976*; *Jordan et al., 1992*; *Thery et al., 2005*; *Harder et al., 2009*; *Kiyomitsu and Cheeseman, 2012*). Therefore, 30 pM was very unlikely to have general or nonspecific effects in the neural progenitors, spindles, or microtubules. Indeed, as described in Results, 30 pM nocodazole did not result in overt perturbations in mitotic morphology, mitotic progression, and general cell and tissue features.

Furthermore, no significant differences were found between spindles in untreated, control (DMSO-only), and 30 pM nocodazole-treated forebrains regarding spindle size, the relative amount of microtubules or the distribution and structure of the fluorescence signal of the microtubules (area, mean fluorescence intensity, and mean standard deviation (SD) of the per-pixel fluorescence intensity values, respectively; *Figure 7B–D*; *Figure 7—source data 1*). A further hypothesis was that, if the spindle, located in the central plane of the cell soma, was somehow affected, the soma diameter at the spindle plane of the cells may be changed in a tissue with high cellular density. This diameter represented also the length of the entire spindle, including some central astrals (*Figure 7E*). No significant differences were found, however, between APs in untreated, control and 30 pM nocodazole-treated forebrains (*Figure 7F*; *Figure 7—source data 1*). In line with our in-organ observations, cell culture experiments have shown that a much higher nocodazole concentration (5 nM) did not disrupt the assembly of a functional bipolar spindle (*Thery et al., 2005*).

The survival of cells in tissue under 30 pM nocodazole was also analysed, by counting the proportion of cleaved caspase 3-immunoreactive cells in cortical wall tissue. The occurrence of positive cells was overall very low, and also similar between control and 30 pM nocodazole (Control 1.49 vs Noc 1.59), showing that this concentration did not result in an apoptosis increase (*Figure 10—figure supplement 1G–I*). Together, these observations strongly suggest that 30 pM nocodazole does not have general or nonspecific effects in the tissue examined.

## Acknowledgements

We thank Services and Facilities of the Max Planck Institute of Molecular Cell Biology and Genetics for outstanding support, notably Jussi Helppi and his team of the Animal Facility, and Jan Peychl and his team of the Light Microscopy Facility. Christiane Haffner provided excellent technical assistance for the cryosection stainings. We thank Jonathon Howard, Stephan Grill, Elena Taverna, YoonJeung Chang, and Miguel Turrero García for helpful discussions and critical reading of the manuscript.

# Additional information

## Funding

| Funder | Grant reference number | Author |
| --- | --- | --- |
| European Molecular Biology Organization | ALTF 1080-2007 | Felipe Mora-Bermúdez |
| Deutsche Forschungsgemeinschaft | SFB 655, A2; TRR 83, Tp6 | Wieland B Huttner |
| European Research Council | 250197 | Wieland B Huttner |
| Deutsche Forschungsgemeinschaft | Center for Regenerative Therapies Dresden | Wieland B Huttner |

| Funder | Grant reference number | Author |
|---|---|---|
| Fonds der Chemischen Industrie | | Wieland B Huttner |

The funders had no role in study design, data collection and interpretation, or the decision to submit the work for publication.

## Author contributions

FM-B, Conceived, designed and performed the experiments, analysed the data and wrote the manuscript; FM, Co-supervised the LGN experiments; WBH, Supervised the project and co-wrote the manuscript

## Ethics

Animal experimentation: All animal studies were conducted in accordance with German animal welfare legislation, and the necessary licenses obtained from the regional Ethical Commission for Animal Experimentation of Dresden, Germany.

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
