## [Decision Letter]

Thank you for sending your work entitled “Specific polar subpopulations of astral microtubules control spindle orientation and symmetric neural stem cell division” for consideration at *eLife.* Your article has been favorably evaluated by Fiona Watt (Senior editor) and 3 reviewers, one of whom is a member of our Board of Reviewing Editors.

The Reviewing editor and the other reviewers discussed their comments before we reached this decision, and the Reviewing editor has assembled the following comments to help you prepare a revised submission.

In this manuscript the authors have examined the different populations of astral microtubules that occur during symmetric versus asymmetric neurogenic divisions of apical radial glial precursors in the embryonic murine cortex. After showing that there is a correlation between the numbers of apical and basal astral microtubules and the spindle orientation variability, they then go on to manipulate the number of apical/basal microtubules and ask whether this influences the spindle orientation variability and, ultimately, symmetric versus neurogenic divisions. The study address a very interesting and timely question, the cellular mechanisms that determine asymmetric versus symmetric stem cell divisions and the paper is well-written and presented. In addition, the experiments, which are largely based upon live imaging approaches, are carefully performed and analyzed. However, the conclusions that the authors draw, while largely convincing, would be greatly strengthened by considering the following issues.

1) The major limitation of this study comes from the assumption that the small amounts of microtubule polymerizing/depolymerizing drugs that are used are specifically affecting the apical/basal astral microtubules. The authors show that features such as spindle area, amount of spindle microtubules, and cell diameter are unchanged. However, they need to ensure that cells progress through mitosis normally and at the same rate as controls. Of course some of the cells complete mitosis, since the authors followed the outcome of divisions after their treatments, but the question is whether drug treatment has reduced and biased the proportion of cells that complete mitosis normally, which would significantly affect their conclusions.

In this regard, two other important parameters that depend upon microtubules and that might be affected are the primary cilium, which acts as a signaling center in these APs, and the elongated bipolar morphology of the APs. The authors should characterize these two parameters to ensure that the drugs are not overtly affecting them.

2) While the authors' data support the proposed model for cells dividing slightly away from the plane of the neuroepithelium, it is less clear how the data could explain cells dividing with their spindle oriented completely perpendicular to the neuroepithelium. Although the authors tend to focus on divisions that deviate only slightly from the plane of the tissue, some more extreme deviations definitely take place in the cortical epithelium, as seen in the quantification of division orientations in Figure 1. It is hard to imagine there would be fewer apical/basal astral microtubules in this situation. How do the authors reconcile this observation with their data?

Minor comments:

1) The authors often refer to “symmetric terminal divisions that produce two neurons”. This statement could be confusing to some readers, as even terminal divisions that generate two neurons could be asymmetric if these neurons are of different types. This should be taken into consideration throughout the text.

2) The LGN immunostaining data would be greatly improved by the addition of a control staining on LGN -/- cortical sections. This is particularly important to increase confidence in the quantification of LGN signal at the apical/central/basal cortex.

3) A recent paper by Juergen Knoblich's group (Juschke et al., Proc Natl Acad Sci U S A. 2014 Jan 21;111(3):1014-9) argues that measuring spindle angles using centrosome labeling in 3D is essential to obtain an accurate representation of division orientation distribution. Here the authors use the cleavage plane as a measure of division orientation, which seems more prone to subjective calls as to where the cleavage line is traced, depending on the quality of the cadherin staining and section angle. The authors might want to consider including a discussion of their method relative to that of Knoblich et al., as this could lead to some controversies.

4) The results in Figure 10 can be interpreted by alternative possibilities such as changes in cellular distribution/location and selective proliferation of certain populations. The authors' previous paper (31) showed that LGN KO changed the distribution of Pax6+ cells but not so much of their number. This apparent inconsistency needs explanation. The authors should add the data of cells in the SVZ (outside of the VZ) in Figure 10. It is also desirable to show the effect of Nocodazole treatment on the % of Tis21+ (neurogenic) cells among APs.

5) Why do the authors detect differences in Tbr1+ cell number in Figure 10, since most Tbr1+ cells (or cells destined to become Tbr1+) are likely to be postmitotic already in E13.5 cortices. Also, 24 h incubation may be too short a time to detect the effects of Nocodazole on progenitor cell fate in the VZ by examining mature neuronal markers in the cortical plate.

---

## [Author Response]

*1) The major limitation of this study comes from the assumption that the small amounts of microtubule polymerizing/depolymerizing drugs that are used are specifically affecting the apical/basal astral microtubules. The authors show that features such as spindle area, amount of spindle microtubules, and cell diameter are unchanged. However, they need to ensure that cells progress through mitosis normally and at the same rate as controls. Of course some of the cells complete mitosis, since the authors followed the outcome of divisions after their treatments, but the question is whether drug treatment has reduced and biased the proportion of cells that complete mitosis normally, which would significantly affect their conclusions*.

The point raised by the reviewers is indeed a very important one. As requested by the reviewers, we have now significantly strengthened the control data showing that cells progress through mitosis normally and at the same rate as controls when exposed to minimal concentrations of the microtubule compounds, i.e. nocodazole and taxol, in the low picomolar range.

In the previous version of the manuscript, we showed that the total number of mitoses counted via phosphohistone 3 (PH3) immunoreactivity in the germinal zones (VZ+SVZ) did not significantly change after incubation with 30 pM nocodazole, compared to the controls with solvent (DMSO) only (Figure 10). This already strongly suggested a normal mitotic progression. We now provide additional data with a second and independent mitosis marker, namely phosphovimentin (pVim), which also shows no change in the total number of mitoses when compared to controls, corroborating a normal mitotic progression (Figure 10—figure supplement 1).

This is also fully consistent with our time-lapse analysis of mitosis progression in live tissue, with high temporal resolution, following 40 cells in 3 independent experiments with minimal nocodazole or taxol (Figure 8 and Figure 9). Compared to controls, this showed indistinguishable rates of normal progression through the mitotic phases, as well as unaffected live mitotic morphology and behaviour. Furthermore, this is in agreement with our detailed subcellular analysis of 40 progenitors incubated with minimal nocodazole or taxol, by immunofluorescence and high-resolution 3D confocal microscopy. This shows mitotic morphologies, as well as a set of quantified key spindle and cell parameters, that are indistinguishable from controls (Figure 6 and Figure 7).

Together, these various sets of controls and the ones mentioned below in our response to the next point raised by the reviewers show that concentrations of microtubule compounds in the low pM range (which is 2-3 orders of magnitude lower that in other spindle studies with ‘low’ concentrations, see Material and Methods) do not significantly affect the normal kinetics and progression through mitosis, nor the general tissue, cellular and subcellular architecture of neural progenitors in the developing cortical wall.

*In this regard, two other important parameters that depend upon microtubules and that might be affected are the primary cilium, which acts as a signaling center in these APs, and the elongated bipolar morphology of the APs. The authors should characterize these two parameters to ensure that the drugs are not overtly affecting them*.

These are great suggestions by the reviewers, and as requested, we have now also analysed the primary cilium as well as the elongated bipolar morphology of APs, incubated with 30 pM nocodazole, and found that those structures are also indistinguishable from the controls incubated with solvent only (DMSO) (new Figure 7—figure supplement 1, respectively).

In addition, we have included another relevant parameter and analysed the cytoarchitecture of the entire E14.5 cortical wall, as revealed by the pattern of microtubules, incubated with 30 pM nocodazole, and found no differences when compared to controls (Figure 7—figure supplement 1). These data further strengthen our conclusion that microtubule compounds in the low picomolar range do not significantly affect the general tissue architecture and the cellular and subcellular morphology of progenitors in the mouse developing cortical wall.

These new data are now described in the manuscript.

*2) While the authors' data support the proposed model for cells dividing slightly away from the plane of the neuroepithelium, it is less clear how the data could explain cells dividing with their spindle oriented completely perpendicular to the neuroepithelium. Although the authors tend to focus on divisions that deviate only slightly from the plane of the tissue, some more extreme deviations definitely take place in the cortical epithelium, as seen in the quantification of division orientations in*
Figure 1*. It is hard to imagine there would be fewer apical/basal astral microtubules in this situation. How do the authors reconcile this observation with their data?*

Divisions of apical progenitors (APs, which undergo mitosis at the ventricular surface) in which the spindle is oriented completely perpendicular to the plane of the neuroepithelium are extremely rare under physiological conditions, as shown by our labs and others. AP divisions typically occur with the spindle oriented parallel to the plane of the neuroepithelium, or with the spindle deviating only slightly from that orientation. Our model of down-regulation of apical/basal astral microtubules primarily applies to these AP divisions (Figure 10).

Cell divisions in which the spindle is oriented completely perpendicular to the plane of the neuroepithelium are mostly observed in basal progenitors (BPs, which undergo mitosis in the subventricular zone). In fact, the cells pointed out by the reviewers in Figure 1 are BPs, not APs. Although BP divisions are not the focus of our study, we do find that apical/basal astrals, and even central astrals, are lower in number in BPs than APs (Figure 2). These data are therefore consistent with our model.

Minor comments:

*1) The authors often refer to “symmetric terminal divisions that produce two neurons”. This statement could be confusing to some readers, as even terminal divisions that generate two neurons could be asymmetric if these neurons are of different types. This should be taken into consideration throughout the text*.

The reviewers are, of course, correct. As suggested, we have rephrased the relevant sentences to clarify this issue.

*2) The LGN immunostaining data would be greatly improved by the addition of a control staining on LGN -/- cortical sections. This is particularly important to increase confidence in the quantification of LGN signal at the apical/central/basal cortex*.

We agree that a control immunostaining of LGN +/– vs. –/– is important. This has been previously published by [31], and we have here used the same mouse line, the same antibodies and similar protocols. To exclude that a different experimenter could produce significantly different results, we provide the requested LGN immunostaining below for the information of the reviewers. In short, the immunofluorescence we present is consistent with that of [31], as it shows the expected enrichment of LGN immunoreactivity in mitotic apical progenitors, in which it is concentrated at the basolateral cell cortex, for LGN +/– E13.5 dorsolateral telencephalon, consistent with the data presented in this manuscript, but no detectable signal in LGN –/– E13.5 dorsolateral telencephalon.Author response image 1.No LGN immunoreactivity in LGN –/– mice. E13.5 LGN+/– (top) and LGN–/– (bottom) mice (as in [31]). 1-μm confocal sections of immunofluorescences for endogenous LGN (red), together with DNA (DAPI, cyan) staining, of coronal sections of the dorsolateral telencephalon. Scale bar = 5 μm.

3) A recent paper by Juergen Knoblich's group (Juschke et al., Proc Natl Acad Sci U S A. 2014 Jan 21;111(3):1014-9.) argues that measuring spindle angles using centrosome labeling in 3D is essential to obtain an accurate representation of division orientation distribution. Here the authors use the cleavage plane as a measure of division orientation, which seems more prone to subjective calls as to where the cleavage line is traced, depending on the quality of the cadherin staining and section angle. The authors might want to consider including a discussion of their method relative to that of Knoblich et al., as this could lead to some controversies.

We agree with the reviewers that the paper by Juschke et al. (2014) is an important paper illustrating the crucial importance of 3D imaging to correctly determine the cleavage plane. In fact, as we describe in detail in Material and Methods and the relevant figure legends, all of our imaging sets, live and fixed, were 3D stacks, and therefore our tracing of the orientation of divisions is valid.

Furthermore, Juschke et al. focus exclusively on the potential difficulties faced when applying the method that uses the centrosome-centrosome axis to calculate cleavage orientation, and they describe important parameters that must be considered when calculating the orientations with this method. The method we used is not dependent on these parameters, as we determined the orientation of divisions by following the actual trajectory of the cleavage plane, in high-resolution and comprehensive 3D stacks of confocal images of dividing progenitors that are in an advanced cytokinesis stage, as we describe in detail in Material and Methods. This does not require the calculation necessary to infer division trajectories from the centrosome-centrosome axis. Therefore, we are confident that our method is equally valid to measure accurate division orientations as the approach of Juschke et al.

*4) The results in*
Figure 10
*can be interpreted by alternative possibilities such as changes in cellular distribution/location and selective proliferation of certain populations. The authors' previous paper (*[31]*) showed that LGN KO changed the distribution of Pax6+ cells but not so much of their number. This apparent inconsistency needs explanation. The authors should add the data of cells in the SVZ (outside of the VZ) in*
Figure 10*. It is also desirable to show the effect of Nocodazole treatment on the % of Tis21+ (neurogenic) cells among APs*.

As requested by the reviewers, we have now included in Figure 10 (new right panel) the Pax6 and Tbr2 data for the SVZ, which show no significant change between control and nocodazole. We have also included a discussion on the differences with previous studies with regards to the distribution and abundance of markers and progenitor types. These differences do not necessarily constitute an inconsistency and could result from the different targets of the perturbations (LGN vs. apical/basal astral microtubules) and from the experimental perturbations themselves (long-term genetic ablation of LGN vs. specific and acute pharmacological down-regulation of apical/basal astrals over 24 h).

Also, as requested by the reviewers, we have performed the experiments to analyse the effect of nocodazole on the proportion of VZ progenitors positive for Tis21:GFP (Figure 10—figure supplement 1). Interestingly, the proportion of Tis21:GFP positive VZ progenitors increases by roughly 10%. This is consistent with similar magnitudes of increase in Tbr2 and of decrease in Pax6 in the VZ (Figure 10 left), as well as with the increase in Tbr1 in the developing cortical plate (Figure 10). Together, these data underscore a general increase in neurogenic progenitors upon incubation with 30 pM nocodazole as compared to controls.

*5) Why do the authors detect differences in Tbr1+ cell number in*
Figure 10*, since most Tbr1+ cells (or cells destined to become Tbr1+) are likely to be postmitotic already in E13.5 cortices. Also, 24 h incubation may be too short a time to detect the effects of Nocodazole on progenitor cell fate in the VZ by examining mature neuronal markers in the cortical plate*.

As we have now included in the Discussion, the increase in Tbr1 (Figure 10) likely results from both, (i) direct neurogenesis from apical progenitors, caused by an increase in asymmetric divisions (Figure 9), and (ii) indirect neurogenesis via basal progenitors, may exhibit a sufficiently short cell cycle to produce neurons in the 24-h incubation performed). This is consistent with the newly added data that show an increase in the neurogenic marker Tis21::GFP in VZ progenitors (see previous point and Figure 10—figure supplement 1).